# CoSpaDi: Compressing LLMS via Calibration-Guided Sparse Dictionary Learning

## Abstract

Post-training compression of large language models (LLMs) often relies on low-rank weight approximations that represent each column of the weight matrix in a shared low-dimensional subspace. This strategy is computationally efficient but the underlying constraint can be overly rigid for heterogeneous projection weights and may incur avoidable accuracy loss. We propose **CoSpaDi** (**Co**mpression via **Spa**rse **Di**ctionary Learning), a training-free framework that replaces low-rank factorization with a structured sparse decomposition in which each weight matrix is represented as a dense dictionary multiplied by a column-sparse coefficient matrix. This yields a union-of-subspaces model: the columns of the weight matrix are represented as linear combinations of different subsets of dictionary atoms, improving expressiveness at a fixed parameter budget. CoSpaDi is *calibration-guided*: using a small calibration set, we optimize the factorization to minimize functional reconstruction error of layer outputs rather than weight-space error. An activation-derived Gram orthonormalization reformulates this data-aware objective into a standard dictionary learning problem on transformed weights, and we support both per-layer compression and cross-layer dictionary sharing within groups of similar projections. Across Llama and Qwen model families, CoSpaDi consistently improves the accuracy–compression and perplexity–compression trade-offs over state-of-the-art SVD-based baselines and strong structured pruning baselines at 20–40% compression ratios. The resulting structured sparsity enables sparse–dense computation and integrates with post-training quantization of the sparse coefficients. CoSpaDi can also be viewed as an intermediate point between matrix factorization and pruning: if the dictionary is fixed to the identity it reduces to pruning in the original weight space, whereas if the coefficient matrix is dense it recovers a standard dense factorization.

## 1 Introduction

Large language models (LLMs) achieve strong performance across diverse tasks, from dialogue and instruction following (Brown et al., 2020; Achiam et al., 2023) to general-purpose reasoning (Touvron et al., 2023; Anil et al., 2023). Their effectiveness stems in part from their Transformer-based architectures that model long-range dependencies with attention (Vaswani et al., 2017; Devlin et al., 2019). At the same time, the scale that enables these capabilities makes LLMs expensive to store and run, creating a practical barrier to deployment on memory- and compute-constrained hardware.

A broad literature addresses post-training compression and acceleration of LLMs, spanning pruning, quantization, knowledge distillation, and structured weight parameterizations such as matrix factorization (Frankle & Carbin, 2019; Dettmers et al., 2022; Hinton et al., 2015; Denton et al., 2014). Among training-free approaches, matrix factorization is particularly attractive because it yields explicit low-parameter surrogates for large projection matrices, and in practice the predominant choice is truncated SVD and its activation-aware variants that use a small calibration set to guide which components to keep and how to scale them (Chen et al., 2021). Cross-layer extensions further reduce overhead by sharing a common low-dimensional subspace across groups of layers (Wang et al., 2025a). Despite strong results, these methods ultimately approximate each matrix within a *single* shared low-dimensional subspace; for heterogeneous Transformer projections, this constraint can be unnecessarily restrictive and motivates richer factorizations.

We study an alternative factorization family that relaxes this constraint. Instead of approximating a matrix with one global low-dimensional basis, we model it using a *union of subspaces* via sparse dictionary learning (Aharon et al., 2006; Elad, 2010): a learned dictionary of atoms is combined with column-sparse coefficients, allowing different columns to be reconstructed from different subsets of atoms. This representation is more flexible than a single shared subspace at the same parameter budget, and it is well aligned with the intuition that different output channels may depend on different latent features.

Building on this idea, we propose **CoSpaDi** (**Co**mpression via **Spa**rse **Di**ctionary Learning), a training-free compression framework for Transformer projections. CoSpaDi learns dictionaries and sparse codes to approximate pretrained weight matrices, and is *data-aware*: from a small calibration set we construct an activation-derived Gram orthonormalization that reformulates functional output reconstruction into a standard dictionary learning problem on a transformed (activation-weighted) weight matrices. The resulting factorization yields structured sparsity that can be paired with post-training quantization of the sparse coefficients, and it naturally supports both *per-layer* compression and *cross-layer* dictionary sharing for groups of related projections.

Conceptually, CoSpaDi also bridges two major post-training compression paradigms: matrix factorization and pruning. Like low-rank methods, it rewrites each weight matrix in a factorized form $W \approx DS$, separating a shared set of atoms from per-column coefficients. At the same time, like pruning methods, its efficiency comes from sparsity, since each output column activates only a small subset of dictionary atoms. These two extremes are recovered as special cases: if the dictionary is the identity, CoSpaDi reduces to pruning in the original weight space; if the coefficient matrix is dense, it becomes a standard dense factorization. This perspective helps position CoSpaDi not merely as an alternative to SVD-based compression, but as a more general structural reparameterization that combines advantages of both viewpoints.

**Contributions.**

- **Beyond low-rank for post-training compression.** We introduce sparse dictionary learning as a new post-training compression paradigm for LLM weight matrices. In contrast to SVD-based methods, which constrain all columns to lie in a single shared low-dimensional subspace, CoSpaDi uses a union-of-subspaces representation in which different columns can activate different subsets of dictionary atoms. This yields a strictly richer factorization class at a matched storage budget and conceptually bridges matrix factorization and pruning.

- **Calibration-guided, training-free optimization.** We integrate sparse dictionary learning with a data-aware objective via activation-derived Gram orthonormalization, yielding a tractable transformed problem that can be optimized efficiently with alternating sparse coding and dictionary updates, without gradient-based fine-tuning.

- **Empirical gains across settings.** Across multiple Llama and Qwen models and a range of compression ratios, CoSpaDi improves the quality–compression trade-off over strong activation-aware SVD baselines (including SVD-LLM (Wang et al., 2025c) in the per-layer setting and Basis Sharing (Wang et al., 2025a) in the grouped setting), and is competitive with recent structured pruning methods (Ma et al., 2023; Shopkhoev et al., 2025).

## 2 Related Work

Post-training compression for Transformers is commonly organized around *what is preserved* (weights, activations, or end-to-end behavior) and *what structure is imposed* (sparsity, low precision, low rank, or shared components). We review the most relevant directions through this lens, and position CoSpaDi as a functional, calibration-guided method that imposes a union-of-subspaces structure via sparse dictionary learning.

**Post-training pruning and practical sparsity.** A central theme in recent LLM compression is to optimize a compressed representation to match layer outputs on a small calibration set, rather than minimizing weight-space error. This perspective appears prominently in one-shot *unstructured* pruning methods such as

SparseGPT (Frantar & Alistarh, 2023) and its simplified activation-driven variants (e.g., Wanda) (Sun et al., 2024), which induce high sparsity with limited degradation by explicitly targeting output preservation. However, unstructured sparsity does not automatically translate into practical *compression* or *speedups*: sparse storage formats typically require additional metadata (e.g., indices in CSR/CSC-like representations), which can substantially reduce effective memory gains at moderate sparsity, and irregular access patterns often make sparse matmuls memory-bound and kernel-limited (Han et al., 2015; Wang, 2020). In practice, inference acceleration usually benefits most from *structured* sparsity patterns and dedicated kernel/hardware support (e.g., 2:4 sparse tensor cores) (Mishra et al., 2021; NVIDIA, 2020). Complementary structured approaches remove or replace higher-level components (blocks/layers) using importance criteria and brief recovery steps (Ma et al., 2023; Shopkhoev et al., 2025). Our work follows the same calibration-guided spirit, but instead of removing weights, we seek a structured factorization that can represent heterogeneous columns more flexibly than a shared low-dimensional subspace.

**Quantization and equivalent transformations.** Quantization reduces memory footprint and can improve throughput, but accuracy at low bit-width often hinges on handling outliers and reshaping activation/weight distributions. Weight-only PTQ methods include vector-wise quantization with outlier routing (Dettmers et al., 2022), Hessian-aware schemes such as GPTQ/OPTQ (Frantar et al., 2023a;b), and activation-aware scaling/outlier protection (Lin et al., 2024; Dettmers et al., 2024). For weight-and-activation quantization, methods such as SmoothQuant (Xiao et al., 2023) and rotation-based transforms (e.g., QuaRot) (Ashkboos et al., 2024) improve quantizability by rebalancing channel ranges or suppressing outliers. These techniques are complementary to structural compression: in our setting, the sparse coefficients produced by CoSpaDi can be quantized post hoc to further reduce memory and potentially improve practical efficiency.

**Low-rank factorization and shared subspaces.** A widely used post-training alternative to pruning is matrix factorization, where each projection is approximated with a low-rank surrogate (Denton et al., 2014). Activation-aware and functional variants include DRONE (Chen et al., 2021), Fisher-weighted reconstruction (FWSVD) (Hsu et al., 2022), and adaptive truncation with activation transforms (ASVD) (Yuan et al., 2023). More recent SVD-LLM methods incorporate truncation-aware whitening and improved allocation strategies across layers (Wang et al., 2025c;b; Qinsi et al., 2025). In parallel, Basis Sharing (Wang et al., 2025a) ties factors across layers to exploit inter-layer redundancy. While effective, these approaches still represent each matrix in a *single* shared subspace (possibly partially shared across layers). Our method targets the same post-training regime, but replaces the single-subspace constraint with a union-of-subspaces representation enabled by sparse coding.

**Dictionary learning and sparse representations.** Dictionary learning and sparse coding have a long history in signal processing and vision (Engan et al., 1999; Aharon et al., 2006; Mairal et al., 2009; Gregor & LeCun, 2010; Elad, 2010), where they provide expressive, parsimonious representations that adaptively select a small subset of atoms per example. In model compression, related ideas appear in structured decompositions such as GroupReduce (Chen et al., 2018) and tensorized factorization schemes (Ma et al., 2019). Recent work has also applied learned dictionaries to other Transformer components, e.g., KV-cache compression via sparse decoding (Kim et al., 2025), and cross-layer sharing formulations specialized to attention (Zhussip et al., 2025), which connects to broader shared-basis approaches (Wang et al., 2025a). In contrast, our focus is a *training-free, calibration-guided* dictionary learning framework for *weight* compression that (i) explicitly optimizes a functional objective through activation-aware Gram orthonormalization, and (ii) supports both per-layer compression and cross-layer dictionary sharing within a unified procedure.

## 3  Method

We first formulate post-training compression objectives in *weight space* and *activation space*. We then reinterpret truncated SVD through the lens of PCA as a "basis times coefficients" model, and introduce sparse dictionary learning as a more expressive union-of-subspaces alternative. Finally, we describe CoSpaDi and its grouped extension, together with compression and complexity estimates.

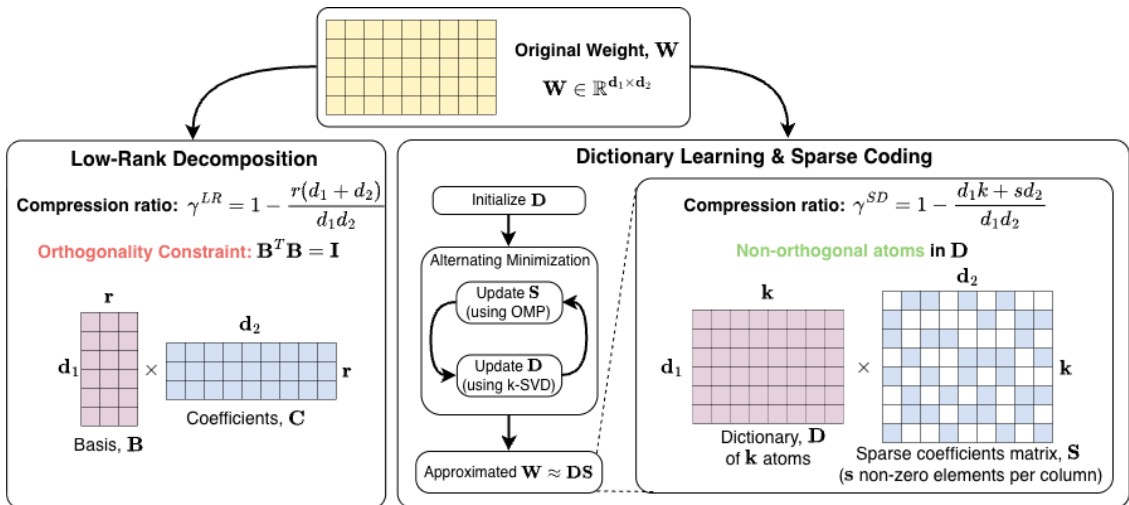

Figure 1: *Left:* low-rank factorization represents $\mathbf{W}$ in a single $r$-dimensional subspace via two dense factors. *Right:* CoSpaDi represents $\mathbf{W}$ as $\mathbf{DS}$ where $\mathbf{D}$ is a dictionary of $k$ atoms and $\mathbf{S}$ is column-sparse (at most $s$ nonzeros per column), yielding a union-of-subspaces model. Dictionaries may be undercomplete ($k < d_1$), complete ($k = d_1$), or overcomplete ($k > d_1$).

### 3.1 Problem formulation: weight-space vs. activation-space objectives

Consider a pretrained linear projection with weight matrix $\mathbf{W} \in \mathbb{R}^{d_1 \times d_2}$ and input $\mathbf{X} \in \mathbb{R}^{N \times d_1}$ (a batch of $N$ calibration activations). The layer output is $\mathbf{Z} = \mathbf{XW} \in \mathbb{R}^{N \times d_2}$. Post-training compression replaces $\mathbf{W}$ by a structured approximation $\tilde{\mathbf{W}}$ that reduces storage and ideally improves inference efficiency.

A common starting point is **weight-space reconstruction**:

$$\tilde{\mathbf{W}}^\star = \underset{\tilde{\mathbf{W}} \in \mathcal{C}}{\arg\min} \|\mathbf{W} - \tilde{\mathbf{W}}\|_F^2, \tag{1}$$

where $\mathcal{C}$ encodes a chosen structure (e.g., rank-$r$, sparsity, quantization, or a parametric factorization).

For LLM projections, however, preserving weights is often a weak proxy for preserving model behavior. A more directly relevant goal is **activation-space (functional) reconstruction** on calibration data:

$$\tilde{\mathbf{W}}^\star = \underset{\tilde{\mathbf{W}} \in \mathcal{C}}{\arg\min} \|\mathbf{XW} - \mathbf{X}\tilde{\mathbf{W}}\|_F^2. \tag{2}$$

Objective (2) is used (implicitly or explicitly) across many calibration-guided post-training methods, and motivates the activation-aware design of CoSpaDi.

### 3.2 Truncated SVD as PCA: basis × coefficients

Low-rank approximation chooses $\mathcal{C} = \{\tilde{\mathbf{W}} : \text{rank}(\tilde{\mathbf{W}}) \leq r\}$ and solves (1):

$$\tilde{\mathbf{W}}^{\text{LR}} = \underset{\text{rank}(\tilde{\mathbf{W}}) \leq r}{\arg\min} \|\mathbf{W} - \tilde{\mathbf{W}}\|_F^2. \tag{3}$$

According to the Eckart–Young–Mirsky theorem (Eckart & Young, 1936), the orthogonal projection to the space of $r$-rank matrices admits an analytical solution. Specifically, if $\mathbf{W}$ admits the singular value decomposition $\mathbf{W} = \mathbf{U\Sigma V}^\mathsf{T}$, with $\mathbf{U} \in \mathbb{R}^{d_1 \times k}$, $\mathbf{\Sigma} \in \mathbb{R}^{k \times k}$ and $\mathbf{V} \in \mathbb{R}^{d_2 \times k}$ with $k = \min(d_1, d_2)$, then the minimizer of Eq. (3) can be expressed as: $\tilde{\mathbf{W}}^{\text{LR}} = \mathbf{U}_r \mathbf{\Sigma}_r \mathbf{V}_r^\mathsf{T}$, where $\mathbf{U}_r \in \mathbb{R}^{d_1 \times r}$ contains the first $r$ left singular vectors, $\mathbf{\Sigma}_r \in \mathbb{R}^{r \times r}$ holds the top-$r$ singular values, and $\mathbf{V}_r \in \mathbb{R}^{d_2 \times r}$ contains the first $r$ right singular vectors of $\mathbf{W}$, respectively.

There is a deep connection between this problem and the principal component analysis (PCA) (Bishop & Nasrabadi, 2006). Specifically, consider the weight matrix $\mathbf{W}$ as a collection of $d_1$-dimensional vectors $\mathbf{W} = \begin{bmatrix} \mathbf{w}_1, & \dots, \mathbf{w}_{d_2} \end{bmatrix}$, where $\boldsymbol{w}_j \in \mathbb{R}^{d_1}$, with $j = 1, \dots, d_2$. We seek to approximate each vector $\mathbf{w}_j$ as a linear combination of basis vectors spanning a lower-dimensional subspace of $\mathbb{R}^{d_1}$. The optimal basis $\mathbf{B} \in \mathbb{R}^{d_1 \times r}$ and coefficients $\mathbf{C} \in \mathbb{R}^{r \times d_2}$ can be found by minimizing the total approximation error:

$$\min_{\mathbf{B}, \mathbf{C}} \ \|\mathbf{W} - \mathbf{BC}\|_F^2 \quad \text{s.t. } \mathbf{B}^\top \mathbf{B} = \mathbf{I}. \tag{4}$$

The optimum is $\mathbf{B}^\star = \mathbf{U}_r$ and $\mathbf{C}^\star = \mathbf{\Sigma}_r \mathbf{V}_r^\top$ (see Appendix A.1 for the proof). Therefore, truncated SVD realizes a *single shared $r$-dimensional subspace*: every column $\mathbf{w}_j$ is expressed in the same basis $\mathbf{B}$ via dense coordinates (the $j$-th column of $\mathbf{C}$) (Figure 1, left).

Although Eq. (3) minimizes reconstruction error in weight space, several works (e.g., (Chen et al., 2021)) note that LLM projection matrices often lack strong intrinsic low-rank structure, so directly approximating $\mathbf{W}$ can lead to noticeable degradation. A common alternative is to optimize *functional* reconstruction on calibration activations, motivated by the hypothesis that layer inputs lie in a lower-dimensional subspace and that preserving the induced outputs is more relevant than preserving weights in isolation. Concretely, this leads to minimizing $\|\mathbf{XW} - \mathbf{X}\tilde{\mathbf{W}}\|_F^2$ for calibration inputs $\mathbf{X}$ (Chen et al., 2021; Hsu et al., 2022; Yuan et al., 2023; Wang et al., 2025c). In this setting, the optimal low-rank approximation can still be obtained in closed form after applying an activation-derived Gram orthonormalization; we provide the full derivation and solution in Appendix A.2.

### 3.3 Sparse dictionary learning: union of subspaces

Dictionary learning keeps the "basis times coefficients" form but changes the modeling assumptions. We replace the orthonormal basis $\mathbf{B}$ by a (generally non-orthogonal) dictionary $\mathbf{D} \in \mathbb{R}^{d_1 \times k}$ and enforce *column sparsity* in the coefficient matrix $\mathbf{S} \in \mathbb{R}^{k \times d_2}$:

$$\min_{\mathbf{D}, \mathbf{S}} \ \|\mathbf{W} - \mathbf{DS}\|_F^2 \quad \text{s.t. } \|\mathbf{s}_j\|_0 \leq s, \ \forall j, \tag{5}$$

where $\mathbf{s}_j$ is the $j$-th column of $\mathbf{S}$, $\|\cdot\|_0$ denotes the $\ell_0$ pseudo-norm, and $s$ indicates the sparsity level.

Compared to low-rank factorization, Eq. (5) differs in two crucial ways: (i) coefficients are sparse, so each column activates only $s$ atoms, and (ii) no orthogonality constraint is imposed on $\mathbf{D}$, allowing more flexible representations. As a result, columns can live in *different* subspaces spanned by different subsets of atoms: $\mathbf{W}$ is represented using a *union of subspaces* model rather than a single shared subspace (Figure 1, right).

### 3.4 CoSpaDi: activation-aware sparse dictionary learning for LLM projections

Motivated by the observation that activations, not projection weights, exhibit low-rank structure, we apply dictionary learning to LLM projection matrices under the activation-space objective (2). Concretely, we choose the constraint set $\mathcal{C} = \{\mathbf{DS} : \mathbf{D} \in \mathbb{R}^{d_1 \times k}, \ \mathbf{S} \in \mathbb{R}^{k \times d_2}, \ \|\mathbf{s}_j\|_0 \leq s\}$ and solve

$$\min_{\mathbf{D}, \mathbf{S}} \ \|\mathbf{XW} - \mathbf{XDS}\|_F^2 \quad \text{s.t. } \|\mathbf{s}_j\|_0 \leq s, \ \forall j. \tag{6}$$

**Gram orthonormalization transform.** To simplify Eq. (6), consider the Gram matrix $\mathbf{G} := \mathbf{X}^\top \mathbf{X} \in \mathbb{R}^{d_1 \times d_1}$. We compute a *Gram orthonormalization* matrix $\mathbf{L}$ from the (uncentered) second moment $\mathbf{G}$ such that

$$\mathbf{Y} := \mathbf{XL}^{-1} \quad \text{satisfies} \quad \mathbf{Y}^\top \mathbf{Y} = \mathbf{I}, \tag{7}$$

i.e., $\mathbf{Y}$ has orthogonal columns. In practice, $\mathbf{L}$ can be obtained via QR factorization of $\mathbf{X}$, or from a factorization of $\mathbf{G}$ (e.g., Cholesky when $\mathbf{G}$ is symmetric positive definite, and an SVD/eigendecomposition-based otherwise). Next, we define the transformed variables $\mathbf{W}_L := \mathbf{LW}$ and $\mathbf{D}_L := \mathbf{LD}$. Using the invariance of the Frobenius norm under left multiplication by a column-orthogonal matrix, Eq. (6) reduces to

$$\min_{\mathbf{D}_L, \mathbf{S}} \ \|\mathbf{W}_L - \mathbf{D}_L \mathbf{S}\|_F^2 \quad \text{s.t. } \|\mathbf{s}_j\|_0 \leq s, \ \forall j. \tag{8}$$

After solving (8), we map back to the original parameterization:

$$\tilde{\mathbf{W}} = \mathbf{D}_a \mathbf{S}, \qquad \mathbf{D}_a = \mathbf{L}^{-1} \mathbf{D}_L. \tag{9}$$

Thus, calibration data enters only through the "whitening" transform $\mathbf{L}$, while the core optimization reduces to standard sparse dictionary learning on $\mathbf{W}_L$.

**Optimization via alternating minimization.** We approximately solve (8) by alternating between sparse coding and dictionary update. Prior to alternating procedure we perform dictionary initialization (while coefficient initialization is also possible with the next step of updating the dictionary) using random permutation set of original columns of projection matrix. Then we perform the following steps in a for loop until stopping criteria is satisfied:

- **Sparse coding.** For each column $\mathbf{w}_{L,j}$ of $\mathbf{W}_L$, we compute a sparse code $\mathbf{s}_j$ with at most $s$ nonzeros by approximately solving $\min_{\mathbf{s}_j} \|\mathbf{w}_{L,j} - \mathbf{D}_L \mathbf{s}_j\|_2^2$ s.t. $\|\mathbf{s}_j\|_0 \leq s$, using orthogonal matching pursuit (OMP) (Tropp & Gilbert, 2007).

- **Dictionary update.** With sparse supports fixed, we update the dictionary. Two common options are: (i) *MOD* (method of optimal directions), which updates all atoms jointly via a global least-squares step (Engan et al., 1999), and (ii) *K-SVD*, which updates atoms sequentially by solving a rank-1 approximation on a restricted residual (Aharon et al., 2006). Concretely, for atom $i$ let $\Omega_i = \{j : (\mathbf{s}_j)_i \neq 0\}$ be the set of columns using atom $i$ and form the restricted residual $\mathbf{E}_i = \left(\mathbf{W}_L - \sum_{p \neq i} \mathbf{d}_p \mathbf{s}_{p,:}\right)_{[:,\Omega_i]}$. K-SVD updates $\mathbf{d}_i$ and $\mathbf{s}_{i,\Omega_i}$ via the top rank-1 component of $\mathbf{E}_i$, which can be computed exactly by SVD or approximated with truncated/randomized low-rank routines (Halko et al., 2011) or power iterations (Golub & Van Loan, 2013). In practice, both steps admit efficient implementations, e.g., Batch-OMP for sparse coding and approximate rank-1 updates for K-SVD, substantially reducing runtime and memory overhead (Rubinstein et al., 2008).

Across our ablations, K-SVD with power-iteration rank-1 updates as well as Batch-OMP provides the strongest performance under a reasonable wall-clock budget. The full procedure is summarized in Appendix A.3.

## 3.5 Grouped / shared-dictionary extension

To exploit inter-layer redundancy, we optionally share a dictionary across a group of projections of the same type. Let $\mathcal{G} = \{\ell_1, \ldots, \ell_L\}$ be a set of layers with weights $\mathbf{W}_\ell \in \mathbb{R}^{d_1 \times d_2}$. We form a grouped matrix by concatenation

$$\mathbf{W}_\mathcal{G} = \left[\mathbf{W}_{\ell_1}, \ldots, \mathbf{W}_{\ell_L}\right] \in \mathbb{R}^{d_1 \times (L d_2)}. \tag{10}$$

To preserve functional fidelity across the group, we stack calibration inputs $\mathbf{X}_\mathcal{G} = \left[\mathbf{X}_{\ell_1}^\top, \ldots, \mathbf{X}_{\ell_L}^\top\right]^\top \in \mathbb{R}^{(LN) \times d_1}$, compute a Gram orthonormalization matrix $\mathbf{L}$ from $\mathbf{X}_\mathcal{G}$, and solve (8) on $\mathbf{W}_{\mathcal{G},L} = \mathbf{L} \mathbf{W}_\mathcal{G}$ to obtain a shared dictionary and block-structured sparse codes. Each layer is then recovered by slicing the corresponding coefficient block.

This approach, inspired by Basis Sharing (Wang et al., 2025a), reduces memory overhead by amortizing the dictionary cost across multiple layers, while preserving activation fidelity through data-aware calibration. Each layer's compressed weights are then recovered by slicing the corresponding coefficient block. Specifically, if $\mathbf{S}_\mathcal{G} \in \mathbb{R}^{k \times (L d_2)}$ are the learned codes, then the codes for layer $\ell_t$ are $\mathbf{S}_{\ell_t} = \mathbf{S}_\mathcal{G}[:, (t-1)d_2 : t d_2]$.

## 3.6 Compression ratio

For low-rank factorization with rank $r$, storing two dense factors costs $r(d_1 + d_2)$ parameters, leading to a compression ratio:

$$\gamma^{\mathrm{LR}} := 1 - \frac{r(d_1 + d_2)}{d_1 d_2}. \tag{11}$$

For CoSpaDi, we store a dense dictionary $\mathbf{D} \in \mathbb{R}^{d_1 \times k}$ and sparse coefficients $\mathbf{S} \in \mathbb{R}^{k \times d_2}$. With $s$ nonzeros per column, the number of stored coefficient values is $sd_2$ (plus indices/mask; see Appendix A.4 for exact accounting). Thus, we can write

$$\gamma^{\mathrm{SD}} := 1 - \frac{\overbrace{d_1 k}^{\text{dict.}} + \overbrace{sd_2}^{\text{values}} + \overbrace{(kd_2)/16}^{\text{mask}}}{d_1 d_2}. \tag{12}$$

Unlike low-rank compression, $\gamma^{\mathrm{SD}}$ depends on two knobs $(k, s)$, enabling additional flexibility at a fixed storage budget. We parameterize the trade-off via $\rho := k/s$, which uniquely determines $(k, s)$ at a target $\gamma^{\mathrm{SD}}$:

$$k = \frac{(1 - \gamma^{\mathrm{SD}}) d_1 d_2}{d_1 + \frac{d_2}{\rho} + \frac{d_2}{16}}, \qquad s = \frac{k}{\rho}. \tag{13}$$

### 3.7 Inference complexity

In terms of computational efficiency, low-rank and dictionary learning exhibit distinct inference-time complexity profiles: the former has a cost of $Nr(d_1 + d_2)$, whereas the latter – when exploiting sparsity and reusing inner products – achieves $Nd_1 K_{\text{active}} + Nsd_2$, potentially yielding superior efficiency under favorable sparsity patterns. Although both methods share identical theoretical complexity under matched compression ratios (Appendix A.5), practical inference latency varies significantly due to factors such as the number of active atoms, indexing overhead, and hardware-specific kernel efficiency. Further details regarding complexity derivations are provided in Appendix A.5.

## 4 Experiments

We evaluate CoSpaDi in two regimes: **per-layer** compression, where each projection matrix is compressed independently, and **grouped (cross-layer)** compression, where a dictionary is shared across a set of layers of the same type. We first present ablations that isolate key design choices (dictionary capacity allocation, data-awareness, packing/quantization, and solver variants), and then report main results for both regimes.

### 4.1 Experimental Setup

**Models and layers.** We evaluate per-layer compression on `LLaMA-3.2-1B`, `Qwen-3-0.6B`, `LLaMA-3-8B`, `Qwen-3-8B` and `Qwen-3-14B`. For *SVD-LLM* we used the original code-base[1] with only first step on compression without extra finetuning. For grouped compression, we follow the *Basis Sharing*[2] protocol and report results on `LLaMA-2-7B`. Unless stated otherwise, we compress all dense linear projections in self-attention (Q/K/V/O) and gated MLP (up/down/gate), while leaving embeddings and the LM head intact.

**Calibration data and metrics.** For calibration we randomly sample 256 sequences of length 1024 from RefinedWeb (Penedo et al., 2023). We report standard zero-shot accuracy (normalized when available) on a suite of benchmarks (PIQA (Bisk et al., 2020), HellaSwag (Zellers et al., 2019), OpenAI LAMBADA (Paperno et al., 2016), ARC-easy/ARC-challenge (Clark et al., 2018), SciQ (Welbl et al., 2017), Race (Lai et al., 2017), MMLU (Hendrycks et al., 2021)) and perplexity on WikiText (Merity et al., 2017) and OpenAI LAMBADA. We used lm-evaluation-harness 0.4.8 (Gao et al., 2024) to ensure reproducibility.

**Compression parameterization.** For low-rank baselines, the target compression ratio (CR) uniquely determines the rank $r$. For CoSpaDi, CR is achieved by choosing the dictionary size $k$ and sparsity $s$ (nonzeros per column), with $\rho := k/s$ controlling how capacity is split between *breadth* (more atoms) and *depth* (more active atoms per column).

---

[1] https://github.com/AIoT-MLSys-Lab/SVD-LLM
[2] https://github.com/TUDa-HWAI/Basis_Sharing

**Baselines.** In the per-layer setting we compare primarily to SVD-LLM (data-aware) (Wang et al., 2025c) and, where relevant, to plain truncated SVD (data-free). We also include comparisons to training-free structured pruning baselines.

We intentionally omit SVD-LLM v2 (Wang et al., 2025b) and Dobi-SVD (Qinsi et al., 2025) from the main baseline set for scope and fairness. Both methods emphasize *dynamic* (layer-wise) compression allocation, which is largely orthogonal to our contribution. Such allocation strategies are not specific to SVD and could be transferred to CoSpaDi by choosing $(k_\ell, s_\ell)$ per layer under a global budget. In addition, SVD-LLM v2 does not provide an official implementation, making it difficult to reproduce calibration and allocation details reliably and to apply them consistently in a dictionary-learning setting. Dobi-SVD further departs from our training-free protocol by requiring backpropagation to optimize the allocation. Its reported gains are also coupled to a quantization-specific "remapping" representation that changes the effective storage accounting. In particular, at moderate compression levels (e.g., CR < 0.5), packing two 16-bit values into a single unit can lead to an effective overparameterization relative to standard bf16 storage. For these reasons, we focus on reproducible training-free baselines with standard storage conventions, and view dynamic allocation and quantization-specific representations as complementary extensions.

**Reproducibility details.** For reproducibility, we provide a consolidated summary of the implementation details used in the main experiments in Appendix A.6, including the solver configuration, dictionary initialization, sparse-coding setup, calibration protocol, storage convention used for the reported compression ratios, and a table of the main hyperparameters.

## 4.2 Ablation Studies

### 4.2.1 Capacity Allocation: the $k/s$ Ratio

At a fixed compression ratio (CR), CoSpaDi admits a family of factorizations parameterized by $\rho := k/s$. Smaller $\rho$ increases per-column expressiveness (larger $s$) but reduces the dictionary size $k$; larger $\rho$ increases $k$ but restricts each column to fewer active atoms. We perform a coarse sweep over $\rho \in [1; 5]$. Figure 2 (LLaMA-3.2-1B) shows that $\rho = 2$ consistently provides the best trade-off across all considered CRs in terms of both average accuracy and perplexity. We therefore fix $\rho = 2$ in all subsequent experiments for simplicity and comparability across models.

Notably, $\rho = 1$ corresponds to the degenerate regime $k = s$, where each column may activate all atoms. In this case, the sparsity constraint becomes inactive and CoSpaDi reduces to a dense two-factor approximation, closely mirroring the "basis times coefficients" view underlying truncated SVD/PCA. This emphasizes that CoSpaDi can be seen as a strict generalization: choosing $\rho > 1$ enforces sparsity and yields a union-of-subspaces model.

### 4.2.2 Data-Free vs. Data-Aware Compression

A core motivation for dictionary learning is that its union-of-subspaces form can be more expressive than a single low-dimensional subspace *regardless of whether calibration is used*. We therefore compare: (i) **data-free** truncated SVD vs. CoSpaDi† (dictionary learning in weight space, without whitening), and (ii) **data-aware** SVD-LLM vs. CoSpaDi (activation-aware via whitening). Table 1 summarizes results on LLaMA-3.2-1B across CR 0.2–0.4. In both regimes, the dictionary-based factorization is consistently stronger than the corresponding low-rank baseline, and activation-aware CoSpaDi yields the best overall trade-off.

### 4.2.3 Dictionary Learning Solver Variants: Exact K-SVD vs. Power-K-SVD vs. MOD

The main bottleneck in the alternating minimization process is the dictionary update using K-SVD as it updates a single atom at a time. To reduce its computational cost we investigate the usage of truncated PCA and power iterations for solving rank-1 problems as well as the well-known Method of Optimal Directions (MOD) which updates in a single step the entire dictionary instead of individual atoms sequentially.

Table 2 reports both quality (Avg. accuracy, WikiText perplexity) and **wall-clock compression time** at a representative 0.2 CR on LLaMA-3.2-1B.Wall-clock is measured for compressing all targeted projection

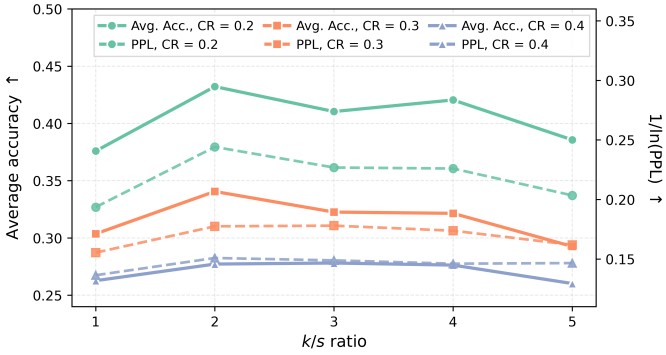 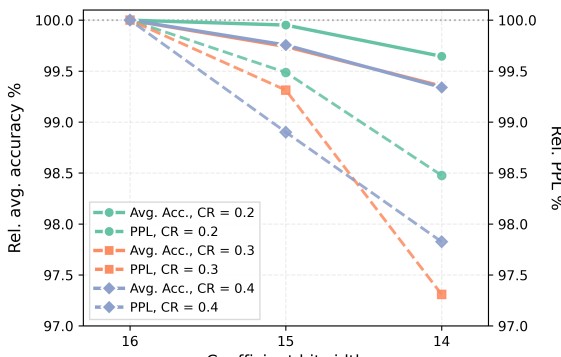

Figure 2: Dual-axis plot showing average accuracy (—— solid lines, left axis) and WikiText perplexity (- - - dashed lines, right axis, inverted logarithmic scale) as functions of $\rho$ for Llama3.2-1B under three compression levels: 0.2, 0.3 and 0.4.

Figure 3: Dual-axis plot showing relative average accuracy (—— solid lines, left axis) and relative WikiText perplexity (- - - dashed lines, right axis) as functions of **S** bitwidth for Llama3-8B under three compression levels: 0.2, 0.3 and 0.4.

Table 1: SDL-based methods comparison vs low-rank counterparts in data-free and data-aware scenarios on Llama3.2-1B at different compression ratios (CR). We denote CoSpaDi† as the proposed method without using calibration data. Best results are provided in **bold**.

| Method | Data-Aware | CR | Accuracy↑ | | | | | | | | | Perplexity↓ | |
| --- | --- | --- | --- | --- | --- | --- | --- | --- | --- | --- | --- | --- | --- |
| | | | PIQA | Hella Swag | LAMBADA | ARC-e | ARC-c | SciQ | Race | MMLU | Avg. | Wiki Text | LAMBADA |
| **Llama3.2 1B** | – | – | 74.5 | 63.7 | 63.0 | 60.5 | 36.2 | 88.3 | 37.8 | 37.0 | 57.6 | 11.6E+00 | 5.7E+00 |
| SVD | ✗ | | 52.0 | 25.7 | 0.0 | 24.7 | 22.1 | 20.0 | 21.4 | 25.4 | 23.9 | 2.9E+06 | 4.6E+06 |
| CoSpaDi† | ✗ | 0.2 | 51.7 | 26.4 | 0.0 | 25.4 | 25.3 | 21.0 | 21.8 | 24.1 | **24.5** | **3.3E+05** | **2.2E+06** |
| SVD-LLM | ✓ | | 62.1 | 36.4 | 24.4 | 36.0 | 25.1 | 64.9 | 29.0 | 23.0 | 37.6 | 1.7E+02 | 1.7E+02 |
| CoSpaDi | ✓ | | 66.1 | 42.9 | 38.4 | 39.9 | 26.0 | 71.6 | 31.7 | 24.8 | **42.7** | **6.4E+01** | **3.5E+01** |
| SVD | ✗ | | 52.3 | 25.6 | 0.0 | 24.2 | 23.5 | 19.5 | 21.9 | 27.0 | 24.3 | 1.1E+06 | 3.9E+06 |
| CoSpaDi† | ✗ | 0.3 | 50.5 | 26.3 | 0.0 | 24.5 | 26.1 | 21.8 | 21.5 | 25.5 | **24.5** | **2.1E+05** | 4.3E+06 |
| SVD-LLM | ✓ | | 55.7 | 30.1 | 9.1 | 30.5 | 21.5 | 45.9 | 25.8 | 23.2 | 30.2 | 5.9E+02 | 2.5E+03 |
| CoSpaDi | ✓ | | 56.9 | 32.4 | 18.2 | 31.9 | 22.1 | 56.7 | 28.0 | 23.1 | **33.7** | **2.9E+02** | **6.6E+02** |
| SVD | ✗ | | 52.8 | 25.9 | 0.0 | 23.9 | 21.3 | 19.9 | 22.2 | 26.9 | 24.1 | 1.2E+06 | 4.2E+06 |
| CoSpaDi† | ✗ | 0.4 | 51.0 | 26.3 | 0.0 | 25.5 | 26.9 | 21.3 | 21.2 | 25.5 | **24.7** | 3.1E+06 | 3.7E+07 |
| SVD-LLM | ✓ | | 51.8 | 27.3 | 1.3 | 26.9 | 22.9 | 32.3 | 24.4 | 23.0 | 26.2 | 1.6E+03 | 3.3E+04 |
| CoSpaDi | ✓ | | 53.5 | 28.2 | 3.8 | 27.8 | 23.0 | 36.9 | 24.0 | 23.1 | **27.5** | **8.0E+02** | **9.2E+03** |

matrices with identical hardware, batching, and stopping criteria. This comparison isolates the practical speed–quality trade-off of the solver choice while keeping the factorization form $(k, s)$ fixed. For our further experiments we utilized K-SVD using power-iterations as it achieves good performance at a reasonable wall-clock time for compression. We provide a more thorough ablation study on the number of power iterations and of the alternating steps in Appendix A.7. To further clarify the reported wall-clock cost, Appendix A.8 provides a concise runtime breakdown of CoSpaDi on a 1B-scale model under our current solver configuration, separating sparse coding, dictionary updates, and miscellaneous overhead. The results show that the dominant cost comes from the sequential K-SVD dictionary-update step.

Table 2: Solver comparison for dictionary update on LLaMA-3.2-1B at 0.2 compression ratio (CR) and fixed $\rho = 2$, 60 alternating minimization iterations. We report accuracies, perplexity as well as wall-clock compression time in minutes using a single A100. Best results are highlighted with **bold**.

| Method | Wall-clock time | Accuracy↑ | | | | | | | | | Perplexity↓ | |
| --- | --- | --- | --- | --- | --- | --- | --- | --- | --- | --- | --- | --- |
| | | PIQA | Hella Swag | LAMBADA | ARC-e | ARC-c | SciQ | Race | MMLU | Avg. | Wiki Text | LAMBADA |
| **Llama3.2 1B** | – | 74.5 | 63.7 | 63.0 | 60.5 | 36.2 | 88.3 | 37.8 | 37.0 | 57.6 | 11.6E+00 | 5.7E+00 |
| MOD | **222.2** | 62.9 | 39.7 | 30.4 | 36.3 | 25.7 | 67.6 | 27.9 | 23.0 | 39.2 | 1.1E+02 | 8.7E+01 |
| K-SVD (PCA) | 902.1 | 64.4 | 39.9 | 33.7 | 40.5 | 25.9 | 72.3 | 31.6 | 24.4 | 41.6 | 9.6E+01 | 6.2E+01 |
| K-SVD (power) | 646.8 | 75.2 | 66.5 | 73.8 | 66.5 | 41.6 | 89.5 | 38.2 | 42.8 | **61.8** | **2.0E+01** | **4.3E+00** |

#### 4.2.4 Quantizing Coefficients and Removing the Mask Term

Sparse factorization introduces metadata overhead for storing nonzero locations. Our default implementation uses a packed binary mask (one bit per entry) plus bf16 values for the $sd_2$ nonzeros. We additionally study **post-training coefficient quantization** by truncating bf16 mantissa bits. Plots on Figure 3 shows that truncating 2 mantissa bits yields negligible degradation across tested CRs.

Beyond robustness, coefficient quantization enables a *simplified storage accounting* in which locations can be treated as fixed by construction (e.g., using a deterministic pattern per column or a reproducible pseudo-random seed), removing the explicit mask term from the compression ratio. Under this view, the effective storage becomes

$$\hat{\gamma}^{\mathrm{SD}} = 1 - \frac{d_1 k + s d_2}{d_1 d_2}. \tag{14}$$

For transparency, Appendix A.9 reports a component-wise memory breakdown separating dictionary storage, coefficient values, and mask/index overhead under both accounting conventions. Unless stated otherwise, we evaluate CoSpaDi with bf16 dictionaries and truncated (14-bit) coefficients, thus providing CR according to Equation 14.

### 4.3 Main Results

#### 4.3.1 Per-Layer Compression

We first compress each projection matrix independently and compare CoSpaDi to SVD-LLM over a range of CRs. Figure 4 summarizes the accuracy–compression and perplexity–compression curves for small models (LLaMA-3.2-1B and Qwen-3-0.6B). For larger models, Table 3 reports results on LLaMA-3-8B, Qwen-3-8B and Qwen-3-14B at CR 0.2–0.4. Across model families and budgets, CoSpaDi consistently improves both average benchmark accuracy and perplexity at matched CR, indicating that the union-of-subspaces factorization better preserves task-relevant directions than a single low-rank basis. We provide additional modern benchmarks in Appendix A.10, including instruction-following and reasoning benchmarks such as IFEval, BBH, MATH, GPQA, MUSR, and MMLU-Pro.

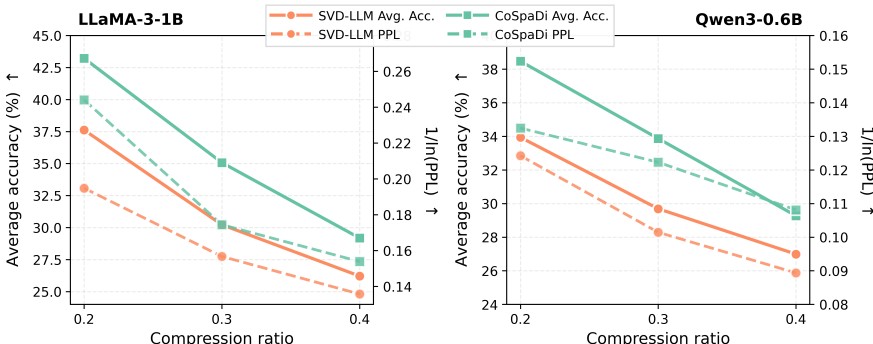

Figure 4: Average benchmark accuracy and WikiText perplexity for LLaMA-3.2-1B and Qwen-3 0.6B using SVD-LLM and CoSpaDi with respect to compression ratio.

#### 4.3.2 Comparison with Pruning

We additionally compare CoSpaDi to recent *structured* pruning approaches on the same models and compression ratios. Specifically, we evaluate against ReplaceMe (Shopkhoev et al., 2025) and LLM-Pruner (Ma et al., 2023), which are representative state-of-the-art methods that reduce parameters by removing or replacing structured components (e.g., blocks/layers) rather than individual weights. Results are reported in Table 4.

We do not include *unstructured* pruning baselines in this comparison because their practical memory savings depend on sparse storage formats and hardware support; at moderate sparsity levels (e.g., $\approx 50\%$), the over-

Table 3: Performance comparison of CoSpaDi vs sota SVD-LLM on Llama3-8B, Qwen3-8B and Qwen3-14B at different compression levels on different benchmarks. Best results are highlighted with **bold**.

| Method | CR | Accuracy↑ | | | | | | | | | Perplexity↓ | |
|---|---|---|---|---|---|---|---|---|---|---|---|---|
| | | PIQA | Hella Swag | LAMBADA | ARC-e | ARC-c | SciQ | Race | MMLU | Avg. | Wiki Text | LAMBADA |
| **Llama3 8B** | – | 80.7 | 79.1 | 75.6 | 77.7 | 53.5 | 93.9 | 40.3 | 62.2 | 70.4 | 7.3E+00 | 3.1E+00 |
| SVD-LLM | 0.2 | 71.1 | 58.4 | 59.3 | 55.5 | 34.0 | 86.4 | 35.5 | 32.6 | 54.1 | 4.1E+01 | 1.1E+01 |
| CoSpaDi | | 75.2 | 66.5 | 73.8 | 66.5 | 41.6 | 89.5 | 38.2 | 42.8 | **61.8** | **2.0E+01** | **4.3E+00** |
| SVD-LLM | 0.3 | 65.8 | 46.4 | 38.1 | 41.9 | 27.7 | 70.0 | 31.8 | 27.2 | 43.6 | 1.5E+02 | 6.1E+01 |
| CoSpaDi | | 70.5 | 56.2 | 61.3 | 54.2 | 33.5 | 85.7 | 36.2 | 32.2 | **53.7** | **4.5E+01** | **9.2E+00** |
| SVD-LLM | 0.4 | 60.3 | 34.5 | 11.4 | 32.4 | 24.5 | 44.2 | 25.7 | 23.1 | 32.0 | 5.5E+02 | 1.3E+03 |
| CoSpaDi | | 63.7 | 41.4 | 30.3 | 39.1 | 26.6 | 68.5 | 30.5 | 25.4 | **40.7** | **1.8E+02** | **1.2E+02** |
| **Qwen3 8B** | – | 77.7 | 74.9 | 64.1 | 80.7 | 56.7 | 95.7 | 40.9 | 73.0 | 70.5 | 1.2E+01 | 4.6E+00 |
| SVD-LLM | 0.2 | 73.8 | 63.9 | 62.2 | 68.7 | 45.7 | 90.1 | 40.5 | 54.7 | 62.5 | 2.1E+01 | 6.4E+00 |
| CoSpaDi | | 76.5 | 68.0 | 65.6 | 72.2 | 48.9 | 93.2 | 40.7 | 60.8 | **65.7** | **1.8E+01** | **4.9E+00** |
| SVD-LLM | 0.3 | 70.4 | 55.2 | 53.8 | 59.3 | 37.1 | 87.2 | 38.4 | 44.8 | 55.8 | 2.7E+01 | 1.1E+01 |
| CoSpaDi | | 72.4 | 60.5 | 62.6 | 63.9 | 41.2 | 88.4 | 39.5 | 51.3 | **59.97** | **2.3E+01** | **6.3E+00** |
| SVD-LLM | 0.4 | 66.3 | 44.6 | 37.9 | 45.0 | 28.1 | 77.3 | 35.3 | 29.1 | 45.4 | 4.3E+01 | 3.6E+01 |
| CoSpaDi | | 68.9 | 49.0 | 49.9 | 49.4 | 29.9 | 82.0 | 36.8 | 36.6 | **50.3** | **3.6E+01** | **1.5E+01** |
| **Qwen3 14B** | – | 79.9 | 78.9 | 67.9 | 82.8 | 60.2 | 96.5 | 43.3 | 77.2 | 73.3 | 1.1E+01 | 3.7E+00 |
| SVD-LLM | 0.2 | 76.2 | 67.6 | 69.1 | 69.8 | 46.8 | 91.0 | 42.9 | 62.5 | 65.8 | 1.8E+01 | 4.1E+00 |
| CoSpaDi | | 77.3 | 71.7 | 72.0 | 73.3 | 51.3 | 91.9 | 43.1 | 65.8 | **68.3** | **1.6E+01** | **3.4E+00** |
| SVD-LLM | 0.3 | 72.0 | 59.7 | 61.5 | 62.6 | 39.3 | 88.9 | 41.5 | 52.0 | 59.7 | 2.3E+01 | 6.5E+00 |
| CoSpaDi | | 74.6 | 64.2 | 69.0 | 65.3 | 43.6 | 90.7 | 42.5 | 58.0 | **63.5** | **2.0E+01** | **4.2E+00** |
| SVD-LLM | 0.4 | 67.4 | 48.7 | 46.0 | 48.5 | 30.7 | 80.6 | 36.7 | 36.8 | 49.4 | 3.5E+01 | 1.8E+01 |
| CoSpaDi | | 69.5 | 52.3 | 57.3 | 53.2 | 33.0 | 84.4 | 37.5 | 43.6 | **53.9** | **3.3E+01** | **8.7E+00** |

head of storing indices and irregular access patterns often diminishes or eliminates effective compression and speedups in standard dense inference pipelines. For completeness, we additionally report in Appendix A.11 a comparison between CoSpaDi and Wanda (Sun et al., 2024) in a 2:4 semi-structured setting. This baseline is particularly relevant from a deployment perspective because it follows a hardware-friendly N:M sparsity pattern and provides a more practical sparse baseline than fully unstructured pruning.

Table 4: Comparison of the proposed CoSpaDi method with state-of-the-art structured pruning methods ReplaceMe Shopkhoev et al. (2025) and LLM-Pruner Ma et al. (2023) on Llama3 8B under different compression ratios. We report accuracy on different benchmarks as well as its average and perplexity. Best results are highlighted with **bold**

| Method | CR | Accuracy↑ | | | | | | | | | Perplexity↓ | |
|---|---|---|---|---|---|---|---|---|---|---|---|---|
| | | PIQA | Hella Swag | LAMBADA | ARC-e | ARC-c | SciQ | Race | MMLU | Avg. | Wiki Text | LAMBADA |
| **Llama3 8B** | | 80.7 | 79.1 | 75.6 | 77.7 | 53.5 | 93.9 | 40.3 | 62.2 | 70.4 | 7.3E+00 | 3.1E+00 |
| ReplaceMe | 0.22 | 73.1 | 65.7 | 42.1 | 65.9 | 43.7 | 86.4 | 35.4 | 51.7 | 58.0 | 3.4E+01 | 2.0E+01 |
| LLM-Pruner | 0.2 | 75.5 | 67.5 | 51.0 | 62.1 | 36.6 | 87.8 | 35.1 | 25.0 | 55.1 | **1.6E+01** | 1.1E+01 |
| CoSpaDi | | 75.2 | 66.5 | 73.8 | 66.5 | 41.6 | 89.5 | 38.2 | 42.8 | **61.8** | 2.0E+01 | **4.3E+00** |
| ReplaceMe | 0.31 | 66.6 | 53.8 | 24.0 | 50.7 | 37.9 | 77.3 | 34.0 | 30.6 | 46.9 | 6.7E+01 | 1.3E+02 |
| LLM-Pruner | 0.3 | 67.3 | 45.1 | 20.9 | 45.4 | 28.8 | 63.4 | 30.1 | 22.9 | 40.5 | **3.8E+01** | 2.2E+02 |
| CoSpaDi | | 70.5 | 56.2 | 61.3 | 54.2 | 33.5 | 85.7 | 36.2 | 32.2 | **53.7** | 4.5E+01 | **9.2E+00** |
| ReplaceMe | 0.41 | 61.7 | 44.3 | 9.8 | 37.4 | 27.5 | 60.4 | 31.6 | 26.4 | 37.4 | 2.3E+02 | 1.8E+03 |
| LLM-Pruner | 0.4 | 50.3 | 25.8 | 1.5 | 26.4 | 25.8 | 28.1 | 21.8 | 23.2 | 25.4 | ∞ | 5.7E+05 |
| CoSpaDi | | 63.7 | 41.4 | 30.3 | 39.1 | 26.6 | 68.5 | 30.5 | 25.4 | **40.7** | **1.8E+02** | **1.2E+02** |

### 4.3.3 Grouped (Cross-Layer) Dictionary Sharing

Next, we evaluate grouped compression where a single dictionary is shared across multiple layers of the same projection type. For a fair and controlled comparison, we adopt the *exact* layer grouping and evaluation

protocol of Basis Sharing, i.e., we share parameters across the same layer sets and report results under the same compression budgets. Table 5 reports results on LLaMA-2-7B across CR 0.2–0.4. We additionally included per-layer counterparts to show the benefits of grouping strategy and exploiting inter-layer redundancies. We include FP16 results (denoted with [*]) to demonstrate that the proposed quantization scheme incurs a negligible performance drop compared to the higher-precision baseline. To facilitate direct comparison with Basis Sharing (Wang et al., 2025a), we report unnormalized accuracies in parentheses alongside the normalized results.

At all budgets, CoSpaDi substantially outperforms Basis Sharing as well as per-layer baselines, suggesting that cross-layer sharing is most effective when paired with sparse codes that allow each column (and each layer) to select its own subset of atoms. We note that CoSpaDi does not rely on this particular grouping and could benefit from more adaptive sharing strategies (e.g., data-driven grouping or partially shared dictionaries), which we leave for future work.

Table 5: Performance comparison of CoSpaDi vs Basis Sharing (Wang et al., 2025a) and per-layer counterparts on Llama2-7B under different compression levels on various benchmarks. Best average accuracy and perplexities are highlighted with **bold**. [*] denotes fp16 coefficient precision and is provided for reference. Unnormalized accuracies are shown in parentheses for direct comparison with (Wang et al., 2025a).

| Method | CR | Accuracy↑ (% (unnorm)) | | | | | | | | | Perplexity↓ | |
| | | PIQA | Hella Swag | LAMBADA | ARC-e | ARC-c | SciQ | Race | MMLU | Avg. | Wiki Text | LAMBADA |
| **Llama2 7B** | – | 78.9 (78.0) | 76.1 (57.3) | 73.8 | 74.2 (76.0) | 45.8 (42.7) | 91.4 (93.9) | 39.7 | 40.8 | 65.1 (62.8) | 8.70 | 3.38 |
| SVD-LLM | | 73.1 (72.1) | 60.3 (44.4) | 63.9 | 55.0 (62.0) | 32.0 (29.1) | 79.5 (89.2) | 36.6 | 25.5 | 53.2 (52.9) | 20.18 | 6.33 |
| CoSpaDi (per-layer)[*] | | 74.3 (73.5) | 64.8 (47.6) | 70.2 | 61.6 (67.3) | 35.3 (34.0) | 85.8 (91.5) | 39.0 | 28.2 | 57.4 (56.4) | 15.82 | 4.53 |
| CoSpaDi (per-layer) | 0.2 | 74.5 (73.6) | 64.5 (47.7) | 70.0 | 61.5 (67.2) | 35.2 (34.0) | 85.4 (91.3) | 39.2 | 28.0 | 57.3 (56.4) | 15.97 | 4.58 |
| Basis Sharing | | 71.1 (70.2) | 60.0 (43.4) | 62.8 | 60.5 (65.5) | 37.8 (33.5) | 85.0 (90.8) | 34.7 | 25.0 | 54.6 (53.2) | 15.17 | 7.03 |
| CoSpaDi (grouped)[*] | | 75.5 (74.3) | 66.5 (48.8) | 71.1 | 68.5 (71.7) | 38.9 (37.5) | 88.7 (91.9) | 38.5 | 27.1 | 59.3 (57.6) | 11.70 | 4.41 |
| CoSpaDi (grouped) | | 75.3 (74.5) | 66.3 (48.8) | 71.1 | 68.1 (71.3) | 39.3 (37.3) | 88.5 (92.2) | 38.5 | 27.0 | **59.3 (57.6)** | **11.73** | **4.42** |
| SVD-LLM | | 68.2 (68.0) | 52.2 (39.4) | 51.6 | 47.3 (53.6) | 28.1 (25.2) | 75.9 (84.9) | 34.6 | 23.5 | 47.7 (47.6) | 33.99 | 13.54 |
| CoSpaDi (per-layer)[*] | | 71.2 (69.9) | 57.4 (42.6) | 62.7 | 53.5 (60.3) | 30.6 (27.9) | 81.6 (87.9) | 36.8 | 26.9 | 52.6 (51.9) | 23.58 | 6.91 |
| CoSpaDi (per-layer) | 0.3 | 71.4 (70.1) | 57.1 (42.4) | 62.7 | 53.2 (60.2) | 31.1 (27.5) | 81.7 (88.3) | 36.3 | 26.7 | 52.5 (51.8) | 23.90 | 6.75 |
| Basis Sharing | | 66.5 (65.4) | 50.3 (37.6) | 53.6 | 54.2 (58.7) | 29.3 (27.2) | 81.4 (86.9) | 32.4 | 23.3 | 48.9 (48.1) | 22.21 | 13.18 |
| CoSpaDi (grouped)[*] | | 70.7 (69.4) | 58.4 (42.4) | 64.5 | 63.6 (67.3) | 35.7 (33.9) | 87.2 (91.0) | 36.1 | 24.1 | 55.0 (53.6) | 15.37 | 6.49 |
| CoSpaDi (grouped) | | 71.0 (68.8) | 58.5 (42.3) | 64.1 | 63.5 (67.4) | 35.5 (33.4) | 87.7 (91.3) | 35.8 | 24.0 | **55.0 (53.4)** | **15.43** | **6.51** |
| SVD-LLM | | 63.3 (62.9) | 42.9 (33.4) | 33.0 | 37.7 (41.9) | 24.7 (21.3) | 70.1 (77.0) | 32.3 | 23.0 | 40.9 (40.6) | 86.50 | 66.91 |
| CoSpaDi (per-layer)[*] | | 66.2 (64.7) | 48.4 (36.9) | 47.2 | 44.3 (49.7) | 26.2 (23.0) | 76.7 (83.7) | 33.4 | 23.4 | 45.7 (45.3) | 49.16 | 20.90 |
| CoSpaDi (per-layer) | 0.4 | 66.0 (65.1) | 47.9 (36.9) | 46.4 | 44.4 (48.9) | 26.5 (22.7) | 76.7 (84.0) | 33.4 | 23.5 | 45.6 (45.1) | 50.46 | 21.50 |
| Basis Sharing | | 60.7 (60.3) | 41.5 (33.0) | 41.0 | 44.6 (49.1) | 26.5 (23.5) | 75.4 (82.8) | 30.1 | 23.2 | 42.9 (42.9) | 39.58 | 36.49 |
| CoSpaDi (grouped)[*] | | 64.6 (63.6) | 48.1 (36.2) | 52.0 | 51.9 (56.7) | 28.9 (25.4) | 80.5 (88.6) | 32.7 | 23.2 | 47.7 (47.3) | 24.99 | 14.71 |
| CoSpaDi (grouped) | | 64.8 (63.4) | 48.0 (36.1) | 51.8 | 51.9 (56.7) | 29.1 (25.6) | 80.5 (88.2) | 32.7 | 23.0 | **47.7 (47.2)** | **25.29** | **14.99** |

### 4.3.4 End-to-end Throughput

Beyond asymptotic complexity, practical deployment depends on realized end-to-end throughput under a fixed runtime stack. To complement the per-layer timing results in Appendix A.5, we additionally measure generation throughput in tokens/s following the evaluation protocol of the SVD-LLM repository, using generation length 1024 and batch sizes 4, 16, and 32.

Two observations are consistent across both models. First, although factorized projection layers reduce the effective compute of the compressed modules, this does not translate into monotonic full-model throughput gains for either SVD or CoSpaDi. This indicates that bottlenecks beyond the decomposed linear layers remain important in the current runtime stack. Second, within the same setup, CoSpaDi remains broadly comparable to SVD-based compression in end-to-end throughput, rather than being systematically worse. These results are consistent with the per-layer inference measurements in Appendix A.5 and suggest that the main practical limitation is not the factorization class itself, but the maturity of sparse runtime support and kernel implementations.

These observations are particularly relevant in light of recent progress in sparse inference at moderate sparsity levels. In our main setting, the default ratio $\rho = k/s = 2$ corresponds to 50% sparsity in the coefficient matrix. This regime is practically meaningful because recent work such as MACKO has reported promising

sparse matrix–vector multiplication speedups already at such moderate sparsity levels, suggesting that the systems outlook for methods such as CoSpaDi may improve as sparse kernels continue to mature (Macko & Boža, 2025). At the same time, translating projection-level FLOP reductions into consistent full-model throughput gains still requires runtime- and kernel-level support, which is outside the main scope of the present work.

Recent results on LLM compression further indicate that sparse inference can already yield practical speedups with existing engines. For example, ELSA reports CPU speedups with DeepSparse and GPU speedups with nm-vllm for sparse inference (Chen et al., 2025; Neural Magic, 2025; 2024). Since CoSpaDi inference involves a dense-times-sparse multiplication, acceleration engines developed for sparse inference are directly relevant to this setting.

## 5 Open Issues and Future Work

**Solver efficiency and scalability.** CoSpaDi uses alternating minimization with OMP for sparse coding and K-SVD-style sequential atom updates. While this optimization procedure is conceptually simple and empirically effective, it can be computationally expensive due to repeated sparse coding and per-atom rank-1 updates, especially for large matrices and aggressive compression. Our runtime breakdown shows that the dominant cost arises from the dictionary-update stage, which reflects the sequential nature of K-SVD. Several acceleration routes are compatible with the current formulation, including more efficient sparse-coding implementations, better batching and parallelization, online or stochastic dictionary learning, and approximate rank-1 updates based on truncated or randomized low-rank routines. Our ablations already indicate that approximate updates based on power iterations provide a favorable accuracy–time trade-off, suggesting that solver efficiency can be improved substantially without changing the underlying CoSpaDi objective.

**Hyperparameter selection and *dynamic* budget allocation.** We parameterize CoSpaDi via $(k, s)$ and $\rho = k/s$, but fix $\rho$ and use uniform budgets for simplicity. Prior low-rank work shows that layer-/type-wise sensitivity can improve quality at a fixed global budget. Adapting dynamic allocation to dictionary learning–e.g., choosing $(k_\ell, s_\ell)$ per layer, projection type, or group based on functional sensitivity–is a promising direction.

**Grouped sharing design choices.** Grouped performance depends on how layers are clustered and which projections share a dictionary. We follow a simple grouping aligned with prior cross-layer sharing, while more adaptive grouping based on activation/weight similarity could yield additional gains.

**Practical speedups depend on kernel support.** CoSpaDi yields structured sparsity exploitable by sparse–dense computation, but end-to-end speedups depend on sparse kernel efficiency, indexing overhead, and reuse of intermediate products. Encouragingly, recent work reports strong acceleration for dense–sparse matmul even at modest sparsity, narrowing the gap between parameter reduction and wall-clock benefits (Macko & Boža, 2025). Realizing consistent speedups will require integration into optimized inference runtimes and hardware-aware kernels for the resulting computation patterns.

**Beyond the current instantiation.** Our goal is to introduce activation-aware sparse dictionary learning as a new post-training compression paradigm beyond single-subspace (low-rank) approximations. Accordingly, we use a direct optimization pipeline. Many advances from low-rank compression–richer calibration objectives, better sensitivity measures, and refined allocation strategies–could be transferred to this setting without changing the core factorization. More broadly, CoSpaDi can be viewed as occupying an intermediate point between matrix factorization and pruning. It inherits the factorized parameterization of low-rank methods through the decomposition $W \approx DS$, while deriving efficiency from sparsity in the coefficient matrix, as in pruning-based approaches. In the limiting cases, the framework reduces to pruning when the dictionary is fixed to the identity, and to dense factorization when the coefficients are not constrained to be sparse. This viewpoint suggests that future advances from both lines of work—better allocation rules from low-rank compression and improved sparse kernels from pruning—can be incorporated into the same dictionary-learning framework.

## 6 Conclusion

We presented CoSpaDi, a training-free LLM compression framework based on sparse dictionary learning. CoSpaDi factorizes projection matrices into a dense dictionary and column-sparse codes, moving from a single shared low-dimensional subspace to a union-of-subspaces representation. Using a small calibration set, an activation-derived Gram orthonormalization converts functional output reconstruction into a tractable dictionary learning objective solved by alternating sparse coding and dictionary updates.

Across multiple Llama and Qwen models, CoSpaDi improves the accuracy–compression and perplexity–compression trade-offs over strong SVD-based baselines in both per-layer and grouped (shared-dictionary) settings at 20–40% compression ratios, and composes naturally with post-training coefficient quantization. More broadly, our results highlight dictionary learning as a flexible paradigm for post-training compression and suggest that many advances from low-rank methods (e.g., dynamic allocation and improved calibration objectives) can be transferred to this richer factorization family.

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

# A    Appendix

## A.1    Derivation of the optimal pair of basis and coefficient matrices

Let us consider the weight matrix $\mathbf{W}$ as a collection of $d_1$-dimensional vectors $\mathbf{W} = \begin{bmatrix} \mathbf{w}_1, & \ldots, \mathbf{w}_{d_2} \end{bmatrix}$, where $\boldsymbol{w}_j \in \mathbb{R}^{d_1}$, with $j = 1, \ldots, d_2$. We seek to approximate each vector $\mathbf{w}_j$ as a linear combination of basis vectors spanning a lower-dimensional subspace of $\mathbb{R}^{d_1}$. The optimal basis and coefficients can be found by minimizing the total approximation error:

$$\mathcal{J} = \sum_{j=1}^{d_2} \left\| \mathbf{w}_j - \sum_{i=1}^{r} c_{i,j} \boldsymbol{b}_i \right\|_2^2 \equiv \| \mathbf{W} - \mathbf{BC} \|_F^2, \tag{15}$$

subject to the orthogonality constraint $\mathbf{B}^\mathsf{T}\mathbf{B} = \mathbf{I}$, where $\mathbf{B} = \begin{bmatrix} \boldsymbol{b}_1, \ldots, \boldsymbol{b}_r \end{bmatrix} \in \mathbb{R}^{d_1 \times r}$ is the basis matrix, $\mathbf{C} \in \mathbb{R}^{r \times d_2}$ is the coefficient matrix with entries $c_{i,j}$, and $\mathbf{I} \in \mathbb{R}^{r \times r}$ is the identity matrix.

To solve this problem, we first reformulate the objective $\mathcal{J}$ into an equivalent form:

$$\mathcal{J} = \mathrm{tr}\left(\mathbf{W}^\mathsf{T}\mathbf{W}\right) - 2\,\mathrm{tr}\left(\mathbf{W}^\mathsf{T}\mathbf{BC}\right) + \mathrm{tr}\left(\mathbf{C}^\mathsf{T}\mathbf{B}^\mathsf{T}\mathbf{BC}\right). \tag{16}$$

Next, we consider the basis matrix $\mathbf{B}$ as fixed and compute the gradient of the objective w.r.t the matrix coefficient $\mathbf{C}$ as

$$\nabla_{\mathbf{C}}\mathcal{J} = 2\mathbf{B}^\mathsf{T}\mathbf{BC} - 2\mathbf{B}^\mathsf{T}\mathbf{W}. \tag{17}$$

Setting the gradient to zero and taking into account the constraint $\mathbf{B}^\mathsf{T}\mathbf{B} = \mathbf{I}$, we can recover the optimum matrix coefficient as: $\mathbf{C} = \mathbf{B}^\mathsf{T}\mathbf{W}$. Now, putting back $\mathbf{C}$ in Eq. (16) we get:

$$\begin{aligned}
\mathcal{J} &= \mathrm{tr}\left(\mathbf{W}^\mathsf{T}\mathbf{W}\right) - 2\,\mathrm{tr}\left(\mathbf{W}^\mathsf{T}\mathbf{BB}^\mathsf{T}\mathbf{W}\right) + \mathrm{tr}\left(\mathbf{W}^\mathsf{T}\mathbf{BB}^\mathsf{T}\mathbf{BB}^\mathsf{T}\mathbf{W}\right) \\
&= \mathrm{tr}\left(\mathbf{W}^\mathsf{T}\mathbf{W}\right) - \mathrm{tr}\left(\mathbf{W}^\mathsf{T}\mathbf{BB}^\mathsf{T}\mathbf{W}\right) \ \text{(from the constraint } \mathbf{B}^\mathsf{T}\mathbf{B} = \mathbf{I}) \\
&= \mathrm{tr}\left(\mathbf{W}^\mathsf{T}\mathbf{W}\right) - \mathrm{tr}\left(\mathbf{B}^\mathsf{T}\mathbf{WW}^\mathsf{T}\mathbf{B}\right).
\end{aligned} \tag{18}$$

Based on this we can recover the optimum basis matrix $\mathbf{B}$ as the maximizer of the constrained problem:

$$\mathbf{B}^\star = \arg\max_{\mathbf{B}} \mathrm{tr}\left(\mathbf{B}^\mathsf{T}\mathbf{WW}^\mathsf{T}\mathbf{B}\right) \ \text{s.t} \ \mathbf{B}^\mathsf{T}\mathbf{B} = \mathbf{I}. \tag{19}$$

The above maximization problem enjoys a closed-form solution Fan (1949), which is fully defined by the eigenvalues of the matrix $\mathbf{P} = \mathbf{WW}^\mathsf{T}$. Specifically, the matrix $\mathbf{P} \in \mathbb{R}^{d_1 \times d_1}$, which is symmetric and positive semi-definite, admits the eigenvalue decomposition $\mathbf{P} = \mathbf{U\Lambda U}^\mathsf{T}$, with $\mathbf{U} \in \mathbb{R}^{d_1 \times d_1}$ holding the eigenvectors

of $\mathbf{P}$ in its columns. Then, the maximizer of Eq. (19) is recovered as $\mathbf{B}^\star = \mathbf{U}_r$ where $\mathbf{U}_r \in \mathbb{R}^{d_1 \times r}$ is a reduced version of $\mathbf{U}$ formed with the $r$ eigenvectors corresponding to the largest eigenvalues of $\mathbf{P}$. A useful observation is that the eigenvectors of $\mathbf{P}$ exactly match the left-singular vectors of $\mathbf{W} \in \mathbb{R}^{d_1 \times d_2}$. Indeed, if $\mathbf{W}$ admits the singular value decomposition $\mathbf{W} = \mathbf{U}\mathbf{\Sigma}\mathbf{V}^\mathsf{T}$, then we have that: $\mathbf{P} = \mathbf{W}\mathbf{W}^\mathsf{T} = \mathbf{U}\mathbf{\Sigma}^2\mathbf{U}^\mathsf{T} \equiv \mathbf{U}\mathbf{\Lambda}\mathbf{U}^\mathsf{T}$, with $\mathbf{\Lambda} = \mathbf{\Sigma}^2$. Therefore, instead of performing the eigenvalue decomposition on $\mathbf{P}$ we can recover $\mathbf{U}$ and consequently $\mathbf{U}_r$ by computing the SVD of $\mathbf{W}$.

Finally, we can compute the optimum coefficient matrix as:

$$\mathbf{C}^\star = (\mathbf{B}^\star)^\mathsf{T}\,\mathbf{W} = \mathbf{U}_r^\mathsf{T}\mathbf{W}$$

$$= \mathbf{U}_r^\mathsf{T} \overbrace{\begin{bmatrix} \mathbf{U}_r & \mathbf{U}_{d-r} \end{bmatrix}}^{\mathbf{U}} \overbrace{\begin{bmatrix} \mathbf{\Sigma}_r & \mathbf{O}_{r\times(d-r)} \\ \mathbf{O}_{(d-r)\times r} & \mathbf{\Sigma}_{d-r} \end{bmatrix}}^{\mathbf{\Sigma}} \overbrace{\begin{bmatrix} \mathbf{V}_r^\mathsf{T} \\ \mathbf{V}_{d-r}^\mathsf{T} \end{bmatrix}}^{\mathbf{V}^\mathsf{T}}$$

$$= \begin{bmatrix} \mathbf{I}_{r\times r} & \mathbf{O}_{r\times(d-r)} \end{bmatrix} \begin{bmatrix} \mathbf{\Sigma}_r\mathbf{V}_r^\mathsf{T} \\ \mathbf{\Sigma}_{d-r}\mathbf{V}_{d-r}^\mathsf{T} \end{bmatrix}$$

$$= \mathbf{\Sigma}_r\mathbf{V}_r^\mathsf{T}, \tag{20}$$

where $\mathbf{V}_r \in \mathbb{R}^{d_2 \times r}$ is a reduced version of $\mathbf{V}$ formed with the $r$ right singular vectors of $\mathbf{W}$ that correspond to its top-$r$ singular values, which are kept in the diagonal matrix $\mathbf{\Sigma}_r \in \mathbb{R}^{r \times r}$.

### A.2   Data-Aware Low-Rank Weight Approximation

While the low-rank approximation of the weights $\mathbf{W}$ has been extensively used for compression tasks, in practice is not well suited to LLMs and it can lead to a severe drop of their performance . Several recent works have suggested that instead of approximating the weights $\mathbf{W}$ with a low-rank matrix, a more efficient strategy is to model the weight activations, $\mathbf{Z} = \mathbf{X}\mathbf{W}$, as low-rank. Here, the matrix $\mathbf{X} = \begin{bmatrix} \mathbf{x}_1 & \dots & \mathbf{x}_N \end{bmatrix}^\mathsf{T} \in \mathbb{R}^{N \times d}$ holds in its rows the $d$-dimensional input vectors $\mathbf{x}_n$ with $n = 1 \dots, N$, which play the role of calibration data. Under this modeling framework, we can approximate the matrix weights $\mathbf{W}$ as the minimizer of the following problem:

$$\tilde{\mathbf{W}}^* = \arg\min_{\tilde{\mathbf{W}}} \left\| \mathbf{X}\mathbf{W} - \mathbf{X}\tilde{\mathbf{W}} \right\|_F \text{ s.t. } \text{rank}\left( \mathbf{X}\tilde{\mathbf{W}} \right) = r. \tag{21}$$

Let us now consider $\mathbf{Y} = \mathbf{X}\mathbf{L}^{-1} \in \mathbb{R}^{N \times d_1}$ to be a semi-orthogonal matrix (column-orthogonal matrix), that is $\mathbf{Y}^\mathsf{T}\mathbf{Y} = \mathbf{I}_{d_1}$, which is obtained by linearly transforming the matrix $\mathbf{X}$ using a non-singular matrix $\mathbf{L} \in \mathbb{R}^{d_1 \times d_1}$. Here we assume that $N \geq d$ and the matrix $\mathbf{X}$ is of full rank. We note that there are different ways we can achieve this column-orthogonalization of $\mathbf{X}$. Among them we can employ the QR/SVD decomposition on $\mathbf{X}$ and the Cholesky/Eigen-value decomposition on $\mathbf{X}^\mathsf{T}\mathbf{X}$ to compute a proper linear transformation $\mathbf{L}$. By using the representation $\mathbf{X} = \mathbf{Y}\mathbf{L}$ we can rewrite the problem of Eq. (21) as:

$$\tilde{\mathbf{W}}^* = \arg\min_{\tilde{\mathbf{W}}} \left\| \mathbf{Y}\mathbf{L}\mathbf{W} - \mathbf{Y}\mathbf{L}\tilde{\mathbf{W}} \right\|_F \text{ s.t. } \text{rank}\left( \mathbf{Y}\mathbf{L}\tilde{\mathbf{W}} \right) = r. \tag{22}$$

To solve the above minimization problem we first note that due to the orthonormal columns of $\mathbf{Y}$, it can be expressed in the equivalent form:

$$\tilde{\mathbf{W}}^* = \arg\min_{\tilde{\mathbf{W}}} \left\| \mathbf{\Delta} - \mathbf{L}\tilde{\mathbf{W}} \right\|_F \text{ s.t. } \text{rank}\left( \mathbf{L}\tilde{\mathbf{W}} \right) = r, \tag{23}$$

where $\mathbf{\Delta} = \mathbf{L}\mathbf{W}$. Next, we introduce the auxiliary matrix $\tilde{\mathbf{\Delta}} = \mathbf{L}\tilde{\mathbf{W}}$ and the problem in Eq. (23) becomes:

$$\tilde{\mathbf{\Delta}}^* = \arg\min_{\tilde{\mathbf{\Delta}}} \left\| \mathbf{\Delta} - \tilde{\mathbf{\Delta}} \right\|_F \text{ s.t. } \text{rank}\left( \tilde{\mathbf{\Delta}} \right) = r, \tag{24}$$

which is the orthogonal projection of $\mathbf{\Delta}$ to the space of $r$-rank matrices. Given that $\tilde{\mathbf{\Delta}}^* = \mathbf{L}\tilde{\mathbf{W}}^*$ and $\mathbf{L}$ is invertible, we can now recover $\tilde{\mathbf{W}}^* = \mathbf{L}^{-1}\tilde{\mathbf{\Delta}}^*$.

To conclude, if $\boldsymbol{\Delta}$ admits the singular value decomposition $\boldsymbol{\Delta} = \mathbf{U}\boldsymbol{\Sigma}\mathbf{V}^\mathsf{T}$, then the optimal $r$-rank approximation of $\mathbf{W}$ that minimizes the loss in Eq. (21) can be written in the form:

$$\tilde{\mathbf{W}} = \mathbf{BC} = \underbrace{\mathbf{L}^{-1}\mathbf{U}_r}_{\mathbf{B}}\underbrace{\boldsymbol{\Sigma}_r\mathbf{V}_r^\mathsf{T}}_{\mathbf{C}}. \tag{25}$$

We note that in this case, unlike the direct weight low-rank approximation, the matrix $\mathbf{B} = \mathbf{L}^{-1}\mathbf{U}_r$ does not correspond to a basis of a subspace of $\mathbb{R}^{d_1}$, since its columns are no longer orthonormal, that is $\mathbf{B}^\mathsf{T}\mathbf{B} = \mathbf{U}_r^\mathsf{T}\left(\mathbf{LL}^\mathsf{T}\right)^{-1}\mathbf{U}_r \neq \mathbf{I}$ .

### A.3 Pseudo Algorithm of the Proposed Method

**Goal.** Given a weight matrix $\mathbf{W} \in \mathbb{R}^{d_1 \times d_2}$ and a small calibration set $\mathbf{X} \in \mathbb{R}^{N \times d_1}$, compute an activation-aware sparse–dictionary factorization $\mathbf{W} \approx \widehat{\mathbf{W}} = \mathbf{DS}$ under a target compression ratio. The procedure consists of whitening the activation objective, alternating sparse coding and dictionary updates on the whitened weights, and a final de-whitening step.

**(1) Calibration and whitening.** Compute an invertible transform $\mathbf{L} \in \mathbb{R}^{d_1 \times d_1}$ (e.g., via QR/SVD of $\mathbf{X}$ or Cholesky of $\mathbf{X}^\mathsf{T}\mathbf{X}$) such that $\mathbf{Y} = \mathbf{XL}^{-1}$ has orthonormal columns ($\mathbf{Y}^\mathsf{T}\mathbf{Y} = \mathbf{I}$). Left-multiply $\mathbf{W}$ to obtain the whitened weights $\mathbf{W}_L = \mathbf{LW}$. Whitening converts the data-aware loss $\|\mathbf{XW} - \mathbf{X}\widehat{\mathbf{W}}\|_F^2$ into a standard Frobenius objective $\|\mathbf{W}_L - \mathbf{D}_L\mathbf{S}\|_F^2$ that is amenable to dictionary learning.

**(2) Initialization.** Initialize the whitened dictionary $\mathbf{D}_\mathbf{L}^{(0)} \in \mathbb{R}^{d_1 \times k}$ (e.g., random permutation of columns of $\mathbf{W}$) and set $\mathbf{S}^{(0)} = 0$. The pair $(k, s)$ is set from the target compression ratio via Eq. 13 optionally using the fixed ratio $\rho = k/s$.

**(3) Alternating minimization.** Repeat for $t = 1, \ldots, T$: *(a) Sparse coding.* For each column $j$, solve

$$\mathbf{s_j^{(t)}} \in \arg\min_{\|\mathbf{s}\|_0 \leq s} \left\|(\mathbf{W}_L)_{:,j} - \mathbf{D_L}^{(t-1)}\mathbf{s}\right\|_2^2,$$

using OMP (greedy selection with orthogonal residual updates) to enforce exactly $s$ nonzeros per column. *(b) Dictionary update.* For each atom $i$, collect its support $\Omega_i = \{j : s_{i,j}^{(t)} \neq 0\}$ and form the residual on those columns:

$$\mathbf{R}_i = \mathbf{W}_L[:, \Omega_i] - \sum_{\ell \neq i} \mathbf{D}_{\mathbf{L},\ell}^{(t-1)} \mathbf{s}_{\ell,\Omega_i}^{(t)}.$$

Update $(\mathbf{D_{L,i}}^{(t)}, \mathbf{s}_{i,\Omega_i}^{(t)})$ by the best rank-1 approximation $\mathbf{R}_i \approx \mathbf{u}\,\sigma\,\mathbf{v}^\top$ (set $/mD_{L,i}^{(t)} \leftarrow \mathbf{u}$, $\mathbf{s}_{i,\Omega_i}^{(t)} \leftarrow \sigma\mathbf{v}^\top$). This preserves the current sparsity pattern while reducing the residual. Iterate atoms sequentially. Stop when the maximum iteration $T$ is reached or when the relative improvement falls below a tolerance.

**(4) De-whitening and packing.** Map the dictionary back to activation space via $\mathbf{D_a} = \mathbf{L}^{-1}\mathbf{D_L^{(T)}}$ and set $\widetilde{\mathbf{W}} = \mathbf{D}_a\mathbf{S}^{(T)}$. For storage, keep $\mathbf{D}_a$ and the $sd_2$ nonzero entries of $\mathbf{S}$ in bf16 along with a packed binary mask $\mathbf{M} \in \{0, 1\}^{k \times d_2}$ for locations (one bit per entry; $\frac{kd_2}{16}$ words). This yields the compression ratio in Appendix A.4 and Eq. 12.

**(5) Inference.** At runtime, apply $\widetilde{\mathbf{W}}$ as matmul$(\mathbf{x}, \mathbf{D}_a\mathbf{S})$ with sparse–dense kernels. Reuse inner products $\langle \mathbf{x}, \mathbf{D}_{a,:,i} \rangle$ across columns to achieve the complexity in Appendix A.5; the number of active atoms controls the practical speedup.

**(6) Cross-layer variant.** For a group of layers $\mathbf{G}$, concatenate weights horizontally $\mathbf{W}_\mathbf{G} = [\mathbf{W}_{\ell_1} \cdots \mathbf{W}_{\ell_\mathbf{L}}]$ and stack calibration batches vertically to form $\mathbf{X}_G$. Compute $\mathbf{L}$ from $\mathbf{X}_G$, run the same alternating procedure on $\mathbf{W}_{L,G} = \mathbf{LW}_G$ to obtain a single shared $\mathbf{D}_a$, and slice the corresponding blocks of $\mathbf{S}_G$ back to per-layer coefficients.

---

**Algorithm 1:** Pseudo algorithm of the proposed CoSpaDi which consists of two steps: (a) sparse coding to compute the coefficients and (b) sequential dictionary update step.

**Input** : $\mathbf{W} \in \mathbb{R}^{d_1 \times d_2}$: weight matrix to compress

$\qquad$ $\mathbf{X} \in \mathbb{R}^{N \times d_1}$: calibration input data ($N$ samples)

$\qquad$ $k$: dictionary size (number of atoms, $k \geq s$)

$\qquad$ $s$: sparsity level (max nonzeros per column in $\mathbf{S}$)

$\qquad$ $T$: number of K-SVD iterations

**Output:** $\mathbf{D}_a \in \mathbb{R}^{d_1 \times k}$: activation-aware dictionary

$\qquad$ $\mathbf{S} \in \mathbb{R}^{k \times d_2}$: sparse coefficient matrix

$\qquad$ $\tilde{\mathbf{W}} = \mathbf{D}_a \mathbf{S}$: compressed weight matrix

Compute $\mathbf{L} \in \mathbb{R}^{d_1 \times d_1}$ such that $\mathbf{Y} = \mathbf{X}\mathbf{L}^{-1}$ satisfies $\mathbf{Y}^\mathsf{T}\mathbf{Y} = \mathbf{I}_{d_1}$;

`% e.g., via QR:` $\mathbf{X} = \mathbf{Q}\mathbf{R} \Rightarrow \mathbf{L} = \mathbf{R}$

`% e.g., via Cholesky:` $\mathbf{X}^\mathsf{T}\mathbf{X} = \mathbf{C}^\mathsf{T}\mathbf{C} \Rightarrow \mathbf{L} = \mathbf{C}$

$\mathbf{W}_L \leftarrow \mathbf{L}\mathbf{W}$;

Initialize $\mathbf{D}_L^{(0)} \in \mathbb{R}^{d_1 \times k}$ with random Gaussian or SVD-based atoms;

Initialize $\mathbf{S}^{(0)} \in \mathbb{R}^{k \times d_2}$ as zero matrix;

**for** $t = 1$ **to** $T$ **do**

$\quad$ **for** $j = 1$ **to** $d_2$ **do**

$\qquad$ $\mathbf{s}_j^{(t)} \leftarrow \arg\min_{\|\mathbf{s}\|_0 \leq s} \left\| \mathbf{W}_{L,j} - \mathbf{D}_L^{(t-1)}\mathbf{s} \right\|_2^2$;

$\qquad$ `% Solve via OMP, LASSO, or thresholding`

$\quad$ **end**

$\quad$ **for** $i = 1$ **to** $k$ **do**

$\qquad$ $\Omega_i \leftarrow \left\{ j \,|\, s_{i,j}^{(t)} \neq 0 \right\}$;

$\qquad$ **if** $\Omega_i \neq \emptyset$ **then**

$\qquad\quad$ $\mathbf{R}_i \leftarrow \mathbf{W}_L[:, \Omega_i] - \sum_{l \neq i} \mathbf{d}_{L,l}^{(t-1)} \mathbf{s}_{l,\Omega_i}^{(t)}$;

$\qquad\quad$ $[\mathbf{u}, \sigma, \mathbf{v}] \leftarrow$ rank-1 SVD of $\mathbf{R}_i$;

$\qquad\quad$ $\mathbf{d}_{L,i}^{(t)} \leftarrow \mathbf{u}$;

$\qquad\quad$ $\mathbf{s}_{i,\Omega_i}^{(t)} \leftarrow \sigma \cdot \mathbf{v}^\mathsf{T}$;

$\qquad$ **end**

$\quad$ **end**

**end**

$\mathbf{D}_a \leftarrow \mathbf{L}^{-1}\mathbf{D}_L^{(T)}$;

$\tilde{\mathbf{W}} \leftarrow \mathbf{D}_a \mathbf{S}^{(T)}$;

**return** $\mathbf{D}_a$, $\mathbf{S}^{(T)}$, $\tilde{\mathbf{W}}$;

---

## A.4 Derivation of the CoSpaDi Compression Ratio

We derive the expression for the compression ratio $\gamma^{\mathrm{SD}}$ of our sparse–dictionary (SD) parameterization. Let $\mathbf{W} \in \mathbb{R}^{d_1 \times d_2}$ be factorized as

$$\mathbf{W} \approx \mathbf{D}\mathbf{S}, \qquad \mathbf{D} \in \mathbb{R}^{d_1 \times k}, \ \mathbf{S} \in \mathbb{R}^{k \times d_2},$$

where each column of $\mathbf{S}$ has exactly $s$ nonzeros (*column-s-sparse*). Throughout, we store real values in `bfloat16` (16 bits) as is common in modern LLMs.

**Dense baseline.** A dense $\mathbf{W}$ requires $d_1 d_2$ `bf16` values.

**Dictionary term.** The dictionary $\mathbf{D}$ stores $d_1 k$ `bf16` values.

**Sparse codes.** Naively, $\mathbf{S}$ would need $k d_2$ values. Since $\mathbf{S}$ is column-$s$-sparse, only $s d_2$ values are stored. For locations, one option is COO: per nonzero we keep a row index and (redundantly) the column index.

Because sparsity is fixed per column, column indices can be omitted; keeping only row indices yields $sd_2$ indices. With 16-bit indices, the total becomes $sd_2$ values $+ sd_2$ indices $= 2sd_2$ 16-bit words. For typical $\rho := k/s = 2$, this equals $kd_2$ words—offering no savings over dense $\mathbf{S}$ storage.

Instead, we use a *bit mask* $\mathbf{M} \in \{0,1\}^{k \times d_2}$ to mark nonzero positions. This requires $kd_2$ bits, i.e., $kd_2/16$ 16-bit words after packing. We then store $sd_2$ `bf16` values for the nonzeros and the packed mask for their positions.

**Total and ratio.**   The SD parameterization thus stores

$$\underbrace{d_1 k}_{\text{dictionary}} + \underbrace{sd_2}_{\text{values}} + \underbrace{\frac{kd_2}{16}}_{\text{mask}}$$

16-bit words. Relative to the dense baseline $d_1 d_2$, the resulting compression ratio is

$$\gamma^{\text{SD}} := 1 - \frac{\overbrace{d_1 k}^{\text{dict. (bf16)}} + \overbrace{sd_2}^{\text{codes (bf16)}} + \overbrace{(kd_2)/16}^{\text{mask (1 bit/entry)}}}{d_1 d_2}.$$

This matches the expression used in the main text and makes explicit the dependence on the two design knobs $(k, s)$.

## A.5   Low-rank and CoSpaDi Inference Complexity

Here we derive the multiplication complexity for the original weight, SVD-compressed weight, and dictionary-learning (k-SVD) compression. We count multiplications only (additions are of the same order). Let $\mathbf{W} \in \mathbb{R}^{d_1 \times d_2}$ be a projection matrix in some layer and $\mathbf{X} \in \mathbb{R}^{N \times d_1}$ be an input feature map; then a dense product $\mathbf{Y} = \mathbf{X}\mathbf{W}$ costs

$$O_{baseline} = Nd_1 d_2. \tag{26}$$

For low-rank, in particular, SVD compression with rank $r$ the projection matrix is approximated with two matrices $\mathbf{W} \approx \mathbf{U}\mathbf{V}$ with $\mathbf{U} \in \mathbb{R}^{d_1 \times r}$ and $\mathbf{V} \in \mathbb{R}^{r \times d_2}$, resulting in the following complexity:

$$O_{LR} = Nd_1 r + Nrd_2 = Nr(d_1 + d_2). \tag{27}$$

Sparse dictionary (SD) learning similarly represents $\mathbf{W} \approx \mathbf{D}\mathbf{S}$ with dictionary $\mathbf{D} \in \mathbb{R}^{d_1 \times k}$ of $k$ atoms and sparse coefficient matrix $\mathbf{S} \in \mathbb{R}^{k \times d_2}$. Omitting sparsity of $\mathbf{S}$ will result in:

$$O_{SD,dense} = Nd_1 k + Nkd_2 = Nk(d_1 + d_2). \tag{28}$$

Taking into account that each column $\mathbf{s}_j$ of $\mathbf{S}$ is $s$-sparse, the $(i,j)$ element of $\mathbf{Y} = \mathbf{X}\mathbf{D}\mathbf{S}$ is

$$y_{i,j} = \sum_{k=1}^{K} \mathbf{S}_{k,j}\langle \mathbf{X}_{i,:}, \mathbf{D}_{:,k}\rangle = \sum_{k \in \mathbf{S}_j} \mathbf{S}_{k,j}\langle \mathbf{X}_{i,:}, \mathbf{D}_{:,k}\rangle, \tag{29}$$

where $S_j = \text{supp}(s_j)$ and $|\mathbf{S}_j| = s$. The overall sparse complexity depends on whether the inner products $\langle \mathbf{X}_{i,:}, \mathbf{D}_{:,k}\rangle$ are reused across columns. With the most efficient way with reuse letting $\mathbf{U} = \bigcup_{j=1}^{d_2} S_j$ and $K_{active} = |U|$ we have:

$$O_{SD,sparse-reuse} = Nd_1 K_{active} + Nsd_2, \qquad s \leq K_{active} \leq \min(K, sd_2). \tag{30}$$

With proposed truncation of 2-bits in mantissa we can omit term for storing indices of nonzero elements in $\mathbf{S}$ resulting in corrected compression ratio:

$$\hat{\gamma}^{SD} = 1 - \frac{d_1 k + sd_2}{d_1 d_2}$$

Table 6: End-to-end throughput (tokens/s) for Llama-3 8B under the evaluation protocol of the SVD-LLM repository.

| Compression | BS=4 | BS=16 | BS=32 |
|---|---|---|---|
| Original | 65.6 | 465.5 | 660.0 |
| SVD (CR=0.5) | 50.9 | 204.0 | 706.4 |
| SVD (CR=0.4) | 49.3 | 184.5 | 496.7 |
| SVD (CR=0.3) | 50.5 | 191.9 | 573.6 |
| SVD (CR=0.2) | 50.1 | 362.1 | 569.6 |
| CoSpaDi (CR=0.5) | 51.5 | 349.2 | 583.7 |
| CoSpaDi (CR=0.4) | 49.8 | 184.4 | 462.8 |
| CoSpaDi (CR=0.3) | 48.9 | 178.3 | 394.9 |
| CoSpaDi (CR=0.2) | 50.0 | 185.7 | 481.0 |

Table 7: End-to-end throughput (tokens/s) for Qwen3-8B under the evaluation protocol of the SVD-LLM repository.

| Compression | BS=4 | BS=16 | BS=32 |
|---|---|---|---|
| Original | 46.9 | 179.3 | 552.4 |
| SVD (CR=0.5) | 37.8 | 304.6 | 504.4 |
| SVD (CR=0.4) | 36.7 | 271.7 | 429.7 |
| SVD (CR=0.3) | 37.3 | 281.2 | 453.6 |
| SVD (CR=0.2) | 36.7 | 263.2 | 401.6 |
| CoSpaDi (CR=0.5) | 37.3 | 249.5 | 404.4 |
| CoSpaDi (CR=0.4) | 36.7 | 267.4 | 404.6 |
| CoSpaDi (CR=0.3) | 36.6 | 275.0 | 430.8 |
| CoSpaDi (CR=0.2) | 37.1 | 286.0 | 455.6 |

The rank for the low-rank decomposition could be estimated from the compression ratio with the following equation:

$$r = \frac{(1 - \gamma^{LR})d_1 d_2}{d_1 + d_2}$$

For sparse dictionary based method we defined $\rho = k/s$ and, thus we can estimate both $k$ and $s$ in the following way:

$$k = \frac{(1 - \hat{\gamma}^{\text{SD}})\, d_1 d_2}{d_1 + \frac{d_2}{\rho}}, \qquad s = \frac{k}{\rho}.$$

Under the same compression ratio for both SVD and CoSpaDi we obtain exactly the same complexity:

$$O_{LR} = Nr(d_1 + d_2) = N(1 - \gamma^{LR})d_1 d_2$$

$$O_{SD} = Nd_1 k + Nsd_2 = N(d_1 k + d_2 \frac{k}{\rho}) = Nk(d_1 + \frac{d_2}{\rho}) = N\frac{(1 - \hat{\gamma}^{\text{SD}})\, d_1 d_2}{d_1 + \frac{d_2}{\rho}}(d_1 + \frac{d_2}{\rho}) = N(1 - \hat{\gamma}^{\text{SD}})\, d_1 d_2$$

$$O_{LR} = O_{SD} = N(1 - \gamma)d_1 d_2. \tag{31}$$

While, theoretical complexity is the same, in practice inference time depends on the sparsity structure ($K_{\text{active}}$), indexing overhead, and kernel efficiency, as can be observed from Figures 5, 6 and 7.

Additionally in Tables 6 and 7 we provide end-to-end throughput using CoSpaDi in comparison with low-rank counterpart showing comparable performance.

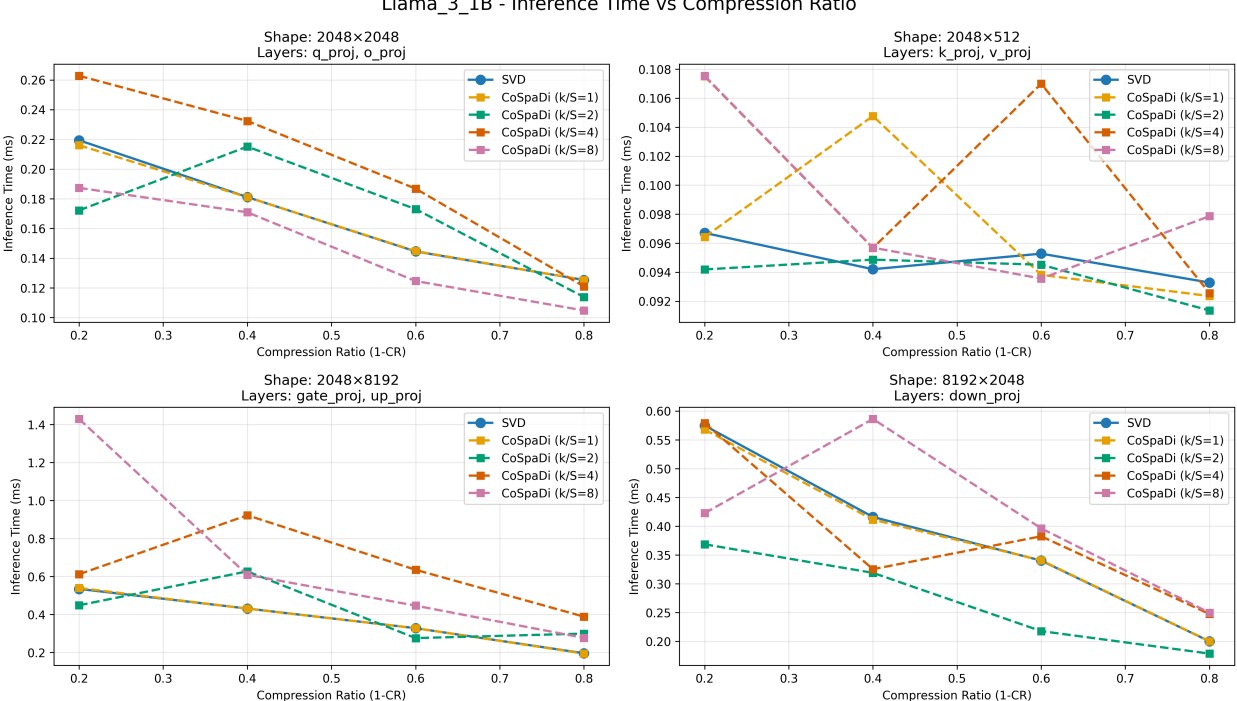

Figure 5: Inference time for different projection layers of Llama3.2 1B for different compression ratios and $k/s$ ratios on A100

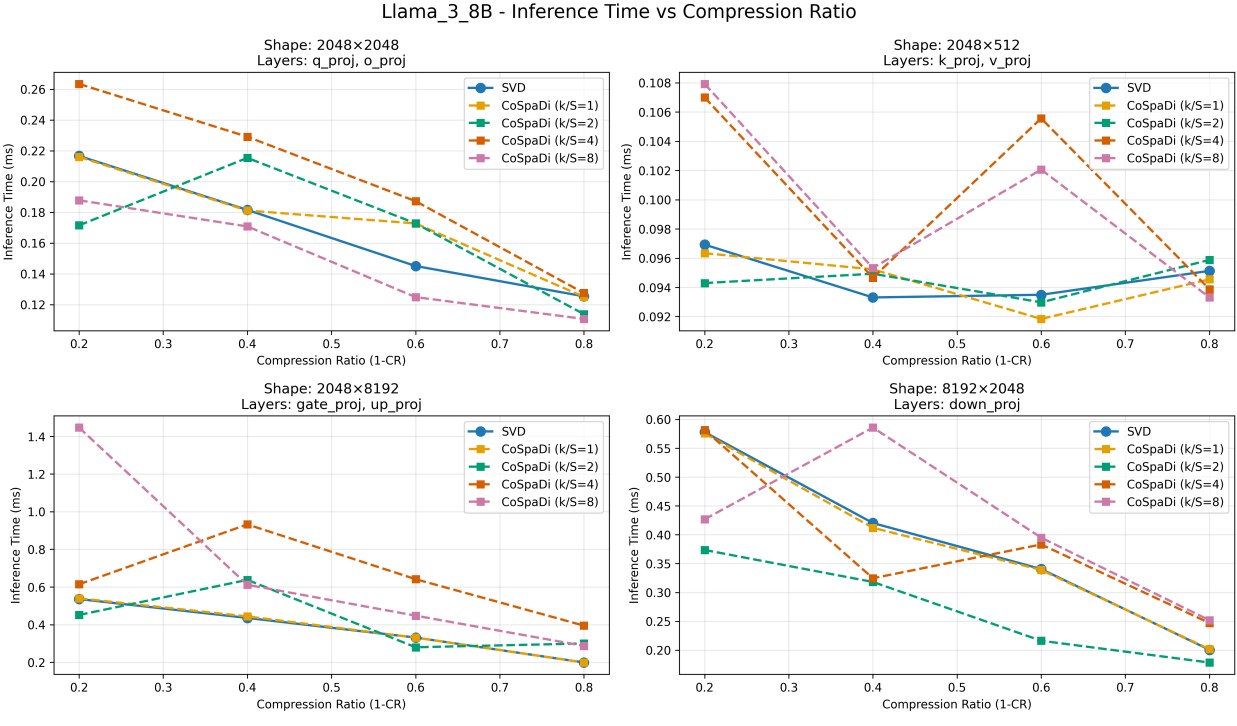

Figure 6: Inference time for different projection layers of Llama3 8B for different compression ratios and $k/s$ ratios on A100

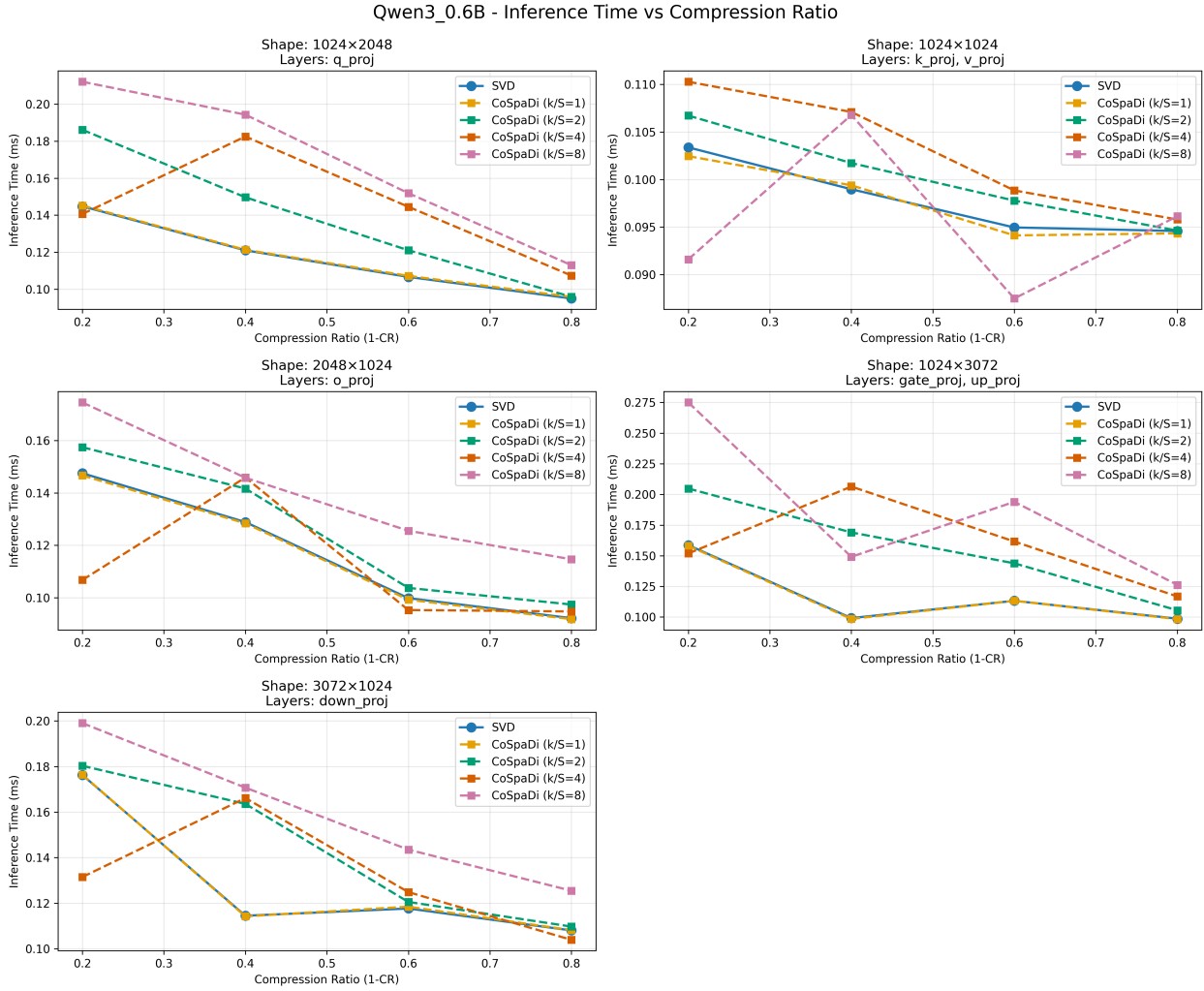

Figure 7: Inference time for different projection layers of Qwen3 0.6B for different compression ratios and $k/s$ ratios on A100

## A.6 Implementation details and reproducibility

This appendix consolidates the main implementation details used in the experiments. Unless otherwise specified, CoSpaDi uses K-SVD with power-iteration-based rank-1 atom updates, with $T = 60$ alternating iterations and 8 power iterations per update. Sparse coding is performed with batched OMP using a mini-batch size of 8192. The dictionary is initialized from a random permutation of columns of the original projection matrix. All experiments use a fixed random seed of 42.

Calibration is performed using 256 sequences of length 1024 sampled from RefinedWeb. Unless stated otherwise, the compressed models use bfloat16 dictionaries and truncated 14-bit coefficients, and the reported compression ratios follow the storage convention described in Section 4.2.4. The summary of the hyperparameters used are listed in Table 8

## A.7 K-SVD and Power Iteration Analysis

We conducted ablation studies to assess the effect of the number of K-SVD iterations and power iterations on performance using Llama3.2-1B with fixed $\rho = 2$. The left plot in Figure 8 shows that average accuracy stabilizes after roughly 50 K-SVD iterations, while perplexity continues to decrease slightly before flattening

Table 8: Main hyperparameters used in the experiments unless otherwise specified. The dictionary size $k$ and sparsity level $s$ are determined from the target compression ratio and the ratio $\rho = k/s$ via Eq. 13.

| Hyperparameter | Value |
|---|---|
| Random seed | 42 |
| K-SVD iterations $T$ | 60 |
| Ratio $\rho = k/s$ | 2.0 |
| Dictionary update solver | power-iteration K-SVD |
| Power iterations | 8 |
| OMP batch size | 8192 |
| Target modules | linear layers in transformer blocks |
| Dictionary size $k$ | determined by CR and $\rho$ via Eq. 13 |
| Sparsity $s$ | determined by CR and $\rho$ via Eq. 13 |
| Dictionary initialization | random columns of $W$ |
| Calibration dataset | RefinedWeb |
| Number of calibration samples | 256 |
| Sequence length | 1024 |
| Storage convention for reported CR | Eq. 14 unless stated otherwise |

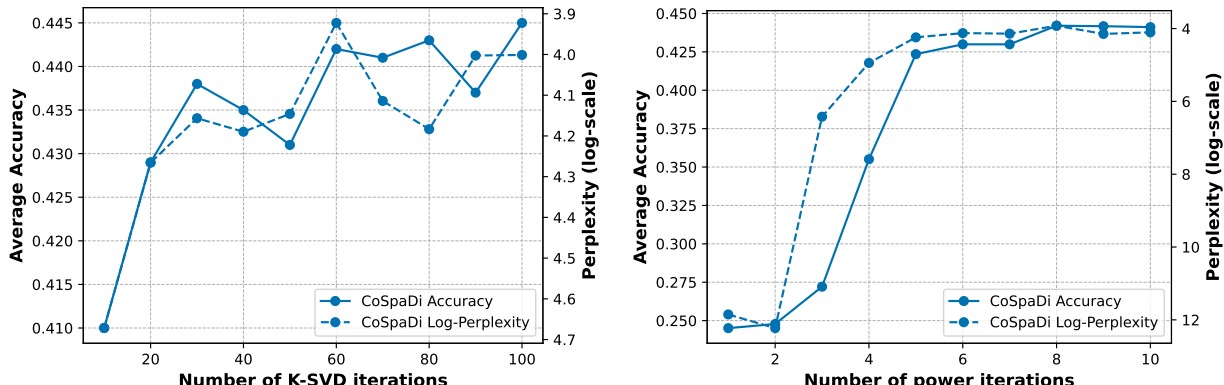

Figure 8: Average benchmark accuracy and WikiText perplexity with respect to the number of K-SVD iterations (left) and the number of power iterations (right) for Llama3.2-1B with $\rho = 2$

out. The right plot of Figure 8 indicates that very few power iterations are sufficient for stable convergence: performance improves sharply up to around 5 iterations, after which additional iterations yield minimal benefit. Based on these results, we fixed the number of K-SVD iterations to 60 and power iterations to 8 in our final experiments, which provides a good balance between accuracy, perplexity, and computational efficiency.

## A.8 Runtime breakdown and scalability discussion

The compression runtime of CoSpaDi is driven by the alternating optimization procedure, which combines OMP-based sparse coding with K-SVD-style dictionary updates. To clarify the contribution of the main components, Table 9 reports a runtime breakdown for end-to-end compression of all targeted projection matrices in a 1B-scale model under the current solver configuration.

The results show that the dominant cost comes from the dictionary-update stage, while sparse coding accounts for a smaller but still substantial fraction of the overall runtime. This behavior is consistent with the optimization structure of CoSpaDi, since K-SVD updates atoms sequentially through repeated rank-1 approximation subproblems.

Table 9: Runtime breakdown of CoSpaDi on a 1B-scale model for end-to-end compression of all targeted projection matrices under the current solver configuration.

| Component | Time (min) | Share (%) |
|---|---|---|
| Sparse coding (OMP) | 178.5 | 27.6 |
| Dictionary update (K-SVD) | 467.6 | 72.3 |
| Misc. overhead | 0.7 | 0.1 |
| Total | 646.8 | 100.0 |

Table 10: Memory footprint breakdown (MB) for storing all compressed linear layers (excluding embeddings, `lm_head`, and normalization layers) under CoSpaDi at different compression ratios.

| Model / Setting | Applied eq. | Quant. coeff. | Dictionary | Coeff. values | Mask / indices | Full total |
|---|---|---|---|---|---|---|
| **Llama-3.2-1B** | – | No | – | – | – | 1856 |
| CoSpaDi, CR=20% | Eq. 12 | No | 990 | 438 | 54 | 1484 |
| | Eq. 14 | Yes | 1018 | 408 | 58 | 1484 |
| CoSpaDi, CR=30% | Eq. 12 | No | 866 | 383 | 47 | 1298 |
| | Eq. 14 | Yes | 890 | 357 | 51 | 1298 |
| CoSpaDi, CR=40% | Eq. 12 | No | 743 | 329 | 41 | 1113 |
| | Eq. 14 | Yes | 763 | 306 | 43 | 1113 |
| **Qwen-3-8B** | – | No | – | – | – | 13248 |
| CoSpaDi, CR=20% | Eq. 12 | No | 7005 | 3192 | 399 | 10596 |
| | Eq. 14 | Yes | 7226 | 2948 | 421 | 10596 |
| CoSpaDi, CR=30% | Eq. 12 | No | 6129 | 2793 | 349 | 9272 |
| | Eq. 14 | Yes | 6322 | 2579 | 368 | 9271 |
| CoSpaDi, CR=40% | Eq. 12 | No | 5253 | 2393 | 299 | 7946 |
| | Eq. 14 | Yes | 5419 | 2211 | 315 | 7947 |

Several acceleration directions are compatible with the current formulation. First, online or stochastic dictionary learning can replace full-batch updates with incremental ones, improving scalability to larger problems (Mairal et al., 2009). Second, implementation-level optimizations such as Batch-OMP and efficient update procedures can substantially reduce wall-clock time in practice (Rubinstein et al., 2008). Third, the rank-1 update step can be approximated using truncated or randomized low-rank routines (Halko et al., 2010), often combined with a small number of power iterations (Golub & Van Loan, 2013), yielding improved accuracy–runtime trade-offs.

These observations suggest that the current compression time is primarily a property of the present solver implementation rather than of the CoSpaDi objective itself. In this work, the focus is on establishing activation-aware sparse dictionary learning as a post-training compression paradigm and evaluating its compression–quality trade-offs; improving compression-time efficiency remains an important direction for future work.

## A.9  Detailed storage breakdown

To make the storage accounting more transparent, we provide in Table 10 a component-wise memory breakdown for CoSpaDi on the main settings. Specifically, we separate the contribution of the dictionary, stored coefficient values, and mask/index overhead. This complements the derivation in Appendix A.4 and clarifies the relation between the full accounting of Eq. (12) and the simplified accounting convention of Eq. (14).

The columns "Applied eq." and "Quant. coeff." indicate whether the target compression ratio is computed using the full accounting of Eq. 12 or the simplified convention of Eq. 14, and whether coefficient truncation is applied. Importantly, the table reports the actual component-wise allocation of storage in each case. Thus, while the relative allocation between dictionary and coefficients changes across the two conventions, the total compressed footprint remains essentially unchanged up to rounding.

For additional intuition, consider a single `down_proj` matrix in Qwen-3 8B with dimensions $d_1 = 4096$ and $d_2 = 12288$.

Under Eq. 12, the dictionary has shape $(4096 \times 3657)$, requiring approximately 28.57 MB. The coefficient matrix has shape $(3657 \times 12288)$ with 1828 nonzero entries per column, yielding approximately 42.84 MB for the stored coefficient values and 5.36 MB for the packed binary mask.

Under Eq. 14, the dictionary has shape $(4096 \times 3932)$, requiring approximately 30.72 MB. The coefficient matrix has shape $(3932 \times 12288)$ with 1966 nonzero entries per column, yielding approximately 40.32 MB for the truncated coefficient values and 5.76 MB for the packed binary mask.

This breakdown makes explicit how much each component contributes to the compressed representation and clarifies that the difference between the two accounting conventions lies primarily in how the storage budget is allocated between dictionary capacity and coefficient precision, rather than in the overall total size.

### A.10 More Benchmarks

We additionally evaluate CoSpaDi and SVD-LLM on the Qwen model family using a set of more recent reasoning-oriented benchmarks. Table 11 reports results for Qwen3-8B and Qwen3-14B across several compression ratios. Overall, CoSpaDi matches or improves upon the SVD-LLM baseline on most benchmarks and settings, indicating that sparse dictionary learning provides a stronger compression–quality trade-off than low-rank factorization beyond the standard evaluation suite.

Table 11: Performance comparison of CoSpaDi vs sota SVD-LLM on Qwen3-8B and Qwen3-14B at different compression levels on different benchmarks. Best results are highlighted with **bold**.

| Model | CR | IFEval (%) | BBH (%) | MATH (%) | GPQA (%) | MUSR (%) | MMLU-Pro (%) |
|---|---|---|---|---|---|---|---|
| Qwen 3 8B | — | 39.2 | 60.9 | 52.6 | 36.2 | 43.1 | 47.7 |
| SVDLLM | 0.2 | 25.5 | 41.0 | 1.1 | 28.4 | 39.8 | 26.3 |
| CoSpaDi | 0.2 | **28.9** | **45.3** | **2.0** | **28.6** | **42.1** | **31.5** |
| SVDLLM | 0.3 | 22.9 | 34.4 | **1.0** | **25.6** | **41.4** | 18.8 |
| CoSpaDi | 0.3 | **25.2** | **38.2** | **1.0** | 24.8 | 38.4 | **22.8** |
| SVDLLM | 0.4 | 22.7 | 30.2 | **0.8** | 23.1 | 37.7 | 11.6 |
| CoSpaDi | 0.4 | **26.1** | **32.7** | **0.8** | **26.0** | **38.1** | **16.7** |
| Qwen 3 14B | — | 43.3 | 63.2 | 53.3 | 38.6 | 40.7 | 53.3 |
| SVDLLM | 0.2 | 25.5 | 50.1 | 1.6 | 29.2 | **41.5** | 32.7 |
| CoSpaDi | 0.2 | **29.6** | **54.2** | **4.2** | **30.6** | 40.7 | **38.9** |
| SVDLLM | 0.4 | 23.0 | 34.3 | 1.1 | **26.4** | **39.7** | 12.8 |
| CoSpaDi | 0.4 | **26.6** | **38.3** | **1.2** | 25.5 | 38.0 | **15.7** |

### A.11 Comparison with semi-structured pruning baseline

To further position CoSpaDi relative to pruning-based compression, we compare it with Wanda (Sun et al., 2024) in a 2:4 semi-structured setting on Llama3 8B. This baseline is particularly relevant from a deployment perspective because it follows a hardware-friendly N:M sparsity pattern.

For the 2:4 baseline, it is important to distinguish nominal sparsity from effective storage compression. Although 2:4 sparsity removes 50% of the weights, the realized memory reduction is smaller once metadata is taken into account. In the standard semi-structured representation, each retained 16-bit weight requires an additional 2 bits of metadata, yielding an effective storage cost of 9 bits per original weight position and therefore an effective compression ratio of $1 - 9/16 \approx 43.8\%$, rather than the naive 50%. We therefore compare Wanda 2:4 to CoSpaDi at the closest matched storage budget.

Results are reported in Table 12. Overall, CoSpaDi remains competitive with pruning-based alternatives and provides the strongest average accuracy at the matched compression ratio. In particular, CoSpaDi outperforms both SVD-LLM and Wanda 2:4 in average accuracy, while also achieving substantially better perplexity than both baselines on LAMBADA and better perplexity than SVD-LLM on WikiText. More broadly, this comparison highlights that CoSpaDi should be viewed not only as an alternative to low-rank

factorization, but also as a sparse reparameterization method that is naturally related to pruning. At the same time, our current formulation is not limited to unstructured sparsity patterns: the same N:M constraints could in principle be imposed directly on the coefficient matrix, allowing each column to activate a hardware-friendly subset of dictionary atoms. We view this as a promising direction for future work.

Table 12: Performance comparison on Llama3 8B between SVD-LLM, CoSpaDi, and Wanda 2:4 at compression ratio 0.4. Best compressed result in each column is highlighted in **bold**.

| Method | CR | Accuracy↑ | | | | | | | | | Perplexity↓ | |
|--------|-----|------|-----------|---------|-------|-------|------|------|------|------|-----------|---------|
| | | **PIQA** | **Hella Swag** | **LAMBADA** | **ARC-e** | **ARC-c** | **SciQ** | **Race** | **MMLU** | **Avg.** | **Wiki Text** | **LAMBADA** |
| Llama3 8B | – | 80.7 | 79.1 | 75.6 | 77.7 | 53.5 | 93.9 | 40.3 | 62.2 | 70.4 | 7.3E+00 | 3.1E+00 |
| SVD-LLM | 0.4 | 60.3 | 34.5 | 11.4 | 32.4 | 24.5 | 44.2 | 25.7 | 23.1 | 32.0 | 5.5E+02 | 1.3E+03 |
| Wanda 2:4 | 0.44 | 62.4 | 39.7 | 0.8 | 41.9 | 25.6 | 74.7 | 26.8 | 23.4 | 36.9 | 2.1E+02 | 5.7E+03 |
| CoSpaDi | 0.4 | 63.7 | 41.4 | 30.3 | 39.1 | 26.6 | 68.5 | 30.5 | 25.4 | **40.7** | **1.8E+02** | **1.2E+02** |

