# OpenReview forum: "CoSpaDi: Compressing LLMs via Calibration-Guided Sparse Dictionary Learning"
_TMLR — Rejected by TMLR_

### Review · Reviewer_53eL · 2026-02-26

**Summary Of Contributions:**

The authors propose a novel post-training compression method for LLMs, named CoSpaDi. This approach employs a Sparse Dictionary Learning strategy to construct a union of subspace representations, thereby replacing conventional low-rank matrix factorization. Finally, the authors convert the activation-aware error into a dictionary learning problem through Gram orthogonalization.
## s1
The motivation of transforming the activation-aware reconstruction objective into a dictionary learning problem via Gram orthogonalization is both clever and intuitive.
## s2
Experimental results demonstrate that the proposed method achieves superior performance.
## w1
The time cost of the proposed method appears to be high, still requiring 600 minutes on a 1B model, which is considerably greater than that of existing quantization methods.
## w2
The experiments only present evaluations on zero-shot selection and perplexity tasks, lacking broader assessments such as mathematical reasoning.

**Audience:**

Yes

**Audience Explanation:**

This work focuses on LLM quantization, which belongs to the field of model quantization and compression.

**Claims And Evidence:**

Yes

**Claims Explanation:**

The authors' motivation is clear and well-founded, and the experimental results demonstrate that the proposed method yields significant improvements. However, I still have concerns regarding some of the findings and evaluation tasks.

**Requested Changes:**

Please refer to W1 and W2

---

> ### Author Response · Authors · 2026-03-04
> **Response to Reviewer 53eL**
>
> We thank the reviewer for his constructive feedback and for recognizing (S1) the intuition behind our Gram-orthogonalization reformulation and (S2) the empirical improvements.
>
> **Positioning (core contribution).**
> Our main goal is to introduce a *conceptually new paradigm* for matrix-factorization-based post-training compression of LLMs: instead of approximating weight matrices via conventional low-rank factorization, we cast activation-aware reconstruction into *sparse dictionary learning* (a union-of-subspaces model), enabled by Gram orthogonalization. We will make this positioning more explicit in the revision.
>
> ## W1: Runtime is high.
> We agree that the current implementation is slower than typical PTQ methods. The reported $\sim$600 minutes corresponds to *end-to-end* compression of all targeted projection matrices for a 1B-scale model under our current solver configuration (single GPU, single process per gpu, fixed iteration budgets). In the revision, we will clarify this scope and add a concise time breakdown to attribute the dominant costs.
>
> **Known acceleration routes and where they already appear in the paper.**
> CoSpaDi is optimized via a standard alternating scheme with OMP-based sparse coding and K-SVD-style sequential atom updates [1]. Since this solver choice largely drives wall-clock time (Table А1), there are well-known and *orthogonal* acceleration avenues that do *not* change the CoSpaDi objective: (i) *online / stochastic dictionary learning* that replaces batch updates with incremental ones for scalability [2], (ii) *implementation-level accelerations* such as Batch-OMP and optimized update strategies for practical speedups [3], and (iii) *approximate / randomized* updates for the inner rank-1 subproblems using truncated or randomized low-rank routines [4] (often combined with a small number of *power iterations* [5]) to obtain favorable accuracy-time trade-offs.
> In addition, a \emph{prominent recent direction} is *randomized/scalable K-SVD* and related randomized dictionary-learning algorithms that explicitly leverage RSVD-style sketching/approximate updates to reduce cost at large scale [6, 7].
>
> Importantly, these directions are *already acknowledged in our submission*: in **Sec. 3.4** we discuss (a) exact vs. approximate computation of the rank-1 update, (b) power-iteration-based approximation, and (c) efficient sparse-coding/update implementations; and in **Sec. 5 (Open Issues and Future Work)** we explicitly flag solver efficiency/scalability as an open issue and point to approximate updates (e.g., power iterations) as a practical trade-off.
> We will strengthen these pointers in the revision and add the above citations for clarity. At the same time, we emphasize that *implementation acceleration is out of scope* for the current paper: our aim here is to establish the *new union-of-subspaces compression paradigm* and demonstrate its accuracy-compression benefits; accelerating the solver is a natural next step building on established dictionary-learning literature.
>
> **Table A1. Runtime breakdown of CoSpaDi on a 1B-scale model (end-to-end compression of all targeted matrices) under our current solver configuration.**
>
> | **Component** | **Time (min)** | **Share (%)** |
> | :--- | ---: | ---: |
> | Sparse coding (OMP) | 178.5 | 27.6 |
> | Dictionary update (K-SVD) | 467.6 | 72.3 |
> | Misc. overhead | 0.7 | 0.1 |
> | **Total** | 646.8 | 100 |
>
> [1] M. Aharon, M. Elad, and A. Bruckstein. K-SVD: An Algorithm for Designing Overcomplete Dictionaries for
> Sparse Representation. TSP, 2006.
>
> [2] J. Mairal, F. Bach, J. Ponce, G. Sapiro, and A. Zisserman. Online Dictionary Learning for Sparse Coding. In
> ICML, 2009.
>
> [3] R. Rubinstein, M. Zibulevsky, and M. Elad. Efficient Implementation of the K-SVD Algorithm using Batch
> Orthogonal Matching Pursuit. Technion, 2008.
>
> [4] N. Halko, P.-G. Martinsson, and J. A. Tropp. Finding Structure with Randomness: Probabilistic Algorithms for
> Constructing Approximate Matrix Decompositions. SIAM Review, 2011.
>
> [5] G. H. Golub and C. F. Van Loan. Matrix Computations. Johns Hopkins University Press, 4th edition, 2013.
>
> [6] G. Wu, Y. Yang, et al. Randomized Algorithms for Large-Scale Dictionary Learning. Neural Networks, 2024.
>
> [7] R. Valentin, et al. DB-KSVD: Scalable Alternating Optimization for Dictionary Learning (Double-Batch KSVD).
> arXiv preprint, 2025.

---

> > ### Author Response · Authors · 2026-03-04
> > **Response to Reviewer 53eL**
> >
> > ## W2: Evaluation breadth.
> > We agree with the reviewer that a broader evaluation should be more prominent in the main paper. Notably, we already report modern reasoning and instruction-following benchmarks in Appendix Table 6 (e.g., BBH, MATH, GPQA, IFEval, MMLU-Pro, MUSR, etc.). In the revision, we will move Table 6 (or a condensed version) into the main text and add a concise discussion linking these results directly to our claims.
> >
> >
> >
> > ## Planned revision changes.
> >
> > - Strengthen the statement of the paper's central contribution: a union-of-subspaces / dictionary-learning paradigm for activation-aware matrix factorization in LLM compression.
> > - Clarify runtime scope and add a short breakdown (Table A1); explicitly mark implementation acceleration as future work (out of scope for current submission).
> > - Promote Appendix Table 6 to the main text and expand the accompanying discussion (no method changes required).

---

> ### Comment · Reviewer_53eL · 2026-03-10
> **Comment**
>
> I appreciate the efforts made by the authors in addressing the questions I raised. Additionally, I have noticed that some other large-model compression methods, such as quantization [1] [2], typically have compression times ranging from 60 to 200 minutes (for models at the 7B scale). Could the authors further discuss the distinctions and connections between quantization-based strategies and the proposed method? (This is not necessarily required to be included in the main text.​)
> ﻿
>
> [1] Billm: pushing the limit of post training quantization for llms. ICML 2024.
> [2] ARB-LLM:Alternating refined binarizations for large language models. ICLR 2025.

---

> > ### Author Response · Authors · 2026-03-11
> > **Response to Reviewer 53eL**
> >
> > **Connection to post-training quantization.**
> >
> > We thank the reviewer for this helpful suggestion. We agree that positioning CoSpaDi relative to post-training quantization (PTQ) methods provides useful context, especially since substantial *offline* compression time is not unique to our approach. In particular, recent ultra-low-bit PTQ methods such as BiLLM and ARB-LLM also rely on nontrivial post-training optimization procedures rather than a purely closed-form transformation [1, 2]. For example, BiLLM reports binarizing a 7B model within about 0.5 hours on a single GPU, which supports the broader point that many practical post-training compression methods incur a one-time offline cost [1]. In this sense, CoSpaDi is similar in spirit to PTQ methods in that compression is performed once offline, after which the compressed model is used repeatedly for inference.
> >
> > At the same time, we view quantization and CoSpaDi as operating on *different and largely complementary compression axes*. Quantization primarily reduces the *numerical precision* used to represent model parameters, while largely preserving the original tensor structure. In contrast, CoSpaDi changes the *parameterization itself*, replacing a dense weight matrix by a learned dictionary and a sparse coefficient matrix. Thus, quantization is fundamentally a bitwidth-reduction strategy, whereas CoSpaDi is a structural reparameterization strategy based on sparse dictionary learning.
> >
> > This distinction also highlights a natural connection between the two families: they are readily *composable*. Since CoSpaDi outputs dictionary and coefficient factors, these factors can themselves be quantized, thereby combining structural compression with reduced numerical precision. Indeed, our paper already includes this type of analysis by showing that CoSpaDi remains effective when the sparse coefficients are truncated to low precision. We therefore view PTQ methods such as BiLLM and ARB-LLM not as alternatives that conflict with CoSpaDi, but rather as complementary techniques that can be layered on top of the proposed factorization.
> >
> > [1] Wei Huang, Yangdong Liu, Haotong Qin, Ying Li, Shiming Zhang, Xianglong Liu, Michele Magno, and Xiaojuan Qi. BiLLM: Pushing the Limit of Post-Training Quantization for LLMs. In Proceedings of ICML, 2024.
> >
> > [2] Zhiteng Li, Xianglong Yan, Tianao Zhang, Haotong Qin, Dong Xie, Jiang Tian, Zhongchao Shi, Linghe Kong, Yulun Zhang, and Xiaokang Yang. ARB-LLM: Alternating Refined Binarizations for Large Language Models. In Proceedings of ICLR, 2025.

---

### Review · Reviewer_X2Ep · 2026-03-06

**Summary Of Contributions:**

The paper proposes CoSpaDi (Compression via Sparse Dictionary Learning), a training-free, post-training weight compression framework for Large Language Models (LLMs). The authors argue that traditional low-rank factorization methods (e.g., truncated SVD) are overly restrictive because they map all columns of a weight matrix into a single, shared low-dimensional subspace. To address this, CoSpaDi utilizes a union-of-subspaces model by factorizing weight matrices into a dense dictionary and a column-sparse coefficient matrix.

To optimize this factorization without fine-tuning, the authors formulate a calibration-guided, activation-space objective. They apply an activation-derived Gram orthonormalization to transform the problem into a standard dictionary learning task, which is then solved using alternating minimization (Orthogonal Matching Pursuit for sparse coding and K-SVD with power iterations for dictionary updates). The method includes a parameterization factor, $\rho=k/s$, to control the trade-off between dictionary capacity and sparsity, and extends to cross-layer dictionary sharing (grouped compression). Empirical evaluations on Llama and Qwen models demonstrate improvements over SVD-LLM, Basis Sharing, and structured pruning baselines at 20-40% compression ratios.

Strengths:
- Replacing single-subspace low-rank factorization with a more expressive sparse dictionary learning paradigm is a highly principled and novel approach to post-training LLM compression.
- The use of activation-derived Gram orthonormalization elegantly bridges functional (activation-space) reconstruction with standard dictionary learning solvers.
- The method shows consistent and convincing gains over strong activation-aware SVD baselines (SVD-LLM) and pruning methods across multiple model scales (0.6B to 14B).
- The framework naturally integrates with post-training quantization, showing robustness to truncating the sparse coefficients down to 14 bits.

Weakness:
- The alternating minimization process, particularly the sequential K-SVD dictionary updates, introduces significant computational overhead during the compression phase compared to the closed-form solutions of SVD.
- While theoretical complexity matches low-rank methods, actual wall-clock speedups rely heavily on specialized sparse-dense matrix multiplication kernels and indexing overhead.
- The method currently uses a uniform compression budget across layers, omitting dynamic, sensitivity-aware allocation strategies that have recently advanced the state-of-the-art in low-rank compression.

**Audience:**

Yes

**Audience Explanation:**

The LLM compression community is highly active, with much of the recent literature focused narrowly on unstructured/structured pruning, quantization, or standard low-rank factorization.

**Claims And Evidence:**

Yes

**Claims Explanation:**

The authors provide extensive zero-shot accuracy and perplexity evaluations across multiple model families (Llama-3, Qwen-3) and scales to support their claims of improved compression-quality trade-offs. They also systematically ablate critical design choices, including the capacity allocation ratio ($\rho$), solver variants (K-SVD, MOD), and the impact of data-awareness (activation vs. weight space), which clearly isolates the source of their empirical gains. The theoretical derivations for the Gram orthonormalization and complexity analyses are mathematically sound and comprehensively documented in the appendices.

However, the claims regarding inference efficiency border on theoretical rather than practical. The authors state that the theoretical complexity equals that of low-rank methods, but they correctly note in the discussion that real-world inference latency varies significantly due to indexing overhead and hardware-specific kernel efficiency. The evidence for practical speedups could be contextualized slightly better in the main text, though it does not invalidate the core accuracy/compression claims.

**Requested Changes:**

Please check the weakness.

---

> ### Author Response · Authors · 2026-03-11
> **Response to Reviewer X2Ep**
>
> We thank the reviewer for the careful reading and for the positive assessment of the paper. We also appreciate the focus on three important points: compression-time overhead, practical inference efficiency, and uniform budget allocation.
>
> **(1) Compression-time overhead.**
> We agree that the current solver is slower at compression time than closed-form SVD-based methods. However, this cost is incurred **once offline**, after which the compressed model is reused for repeated inference, so in practice it is less critical than inference-time overhead. This aspect is already analyzed in the paper: Table 2 reports wall-clock compression time under identical hardware and stopping criteria, and shows that the overhead can be reduced substantially by replacing exact sequential K-SVD updates with approximate dictionary-update procedures, including truncated K-SVD, power-iteration-based rank-1 updates, and MOD, together with Batch-OMP for efficient sparse coding [1, 2, 3]. More broadly, further gains should be possible via more parallel/block dictionary updates and online or stochastic dictionary-learning variants [4]. We therefore view compression cost as an important but manageable engineering limitation rather than a conceptual weakness of the paradigm.
>
> **(2) Practical efficiency vs. theoretical complexity.**
> We agree that asymptotic complexity and realized latency should be distinguished carefully. This distinction is already made in the paper. In Section 3.7 and Appendix A.5, we show that under matched compression ratios, low-rank factorization and CoSpaDi have the same asymptotic multiplication complexity. At the same time, we explicitly note that practical latency depends on the sparsity pattern, indexing overhead, intermediate reuse, and hardware/kernel support; Figures 5-7 therefore report measured inference-time behavior on A100 for different matrices and settings.
>
> We would also like to emphasize an important conceptual point: CoSpaDi can be interpreted as *sparsifying weights in a learned dictionary-induced subspace*. This places it between matrix factorization and unstructured pruning. At one extreme, if the dictionary is the identity, the method reduces to pruning in the original weight space; at the other extreme, if the coefficient matrix is dense, it becomes a standard dense factorization. In this sense, CoSpaDi forms a bridge between structured matrix factorization and unstructured pruning.
>
> This perspective is also practically relevant. Since sparsity in CoSpaDi appears in the coefficient matrix, acceleration techniques developed for unstructured sparse inference are directly relevant here as well. For example, recent work on MACKO shows that sparse matrix-vector multiplication can deliver *actual* speedups even at modest sparsity levels typical for LLM pruning, including 50 % sparsity [5]. This is especially relevant to our setup, since the used setting $k/s=2$ for main results corresponds exactly to 50 % sparsity in the coefficient matrix. The public MACKO implementation further supports this practical argument by demonstrating end-to-end gains for LLM inference in this regime [6]. More generally, well-known post-training unstructured pruning methods such as SparseGPT and Wanda already operate in the regime where sparse inference kernels matter [7, 8]. We therefore believe CoSpaDi should be viewed not only as a competitor to low-rank compression, but also as a framework that inherits the growing systems ecosystem around sparse computation.

---

> ### Author Response · Authors · 2026-03-11
> **Response to Reviewer X2Ep**
>
> **(3) Uniform budgets vs. dynamic allocation.**
> We fully agree that dynamic, sensitivity-aware allocation is a promising extension. The paper already discusses this explicitly in Section 4.1 when explaining the role of methods such as SVD-LLM V2 [9] and Dobi-SVD [10]. Our point is not that dynamic allocation is unimportant, but that it is largely orthogonal to the core contribution of this paper: the comparison between single-subspace low-rank compression and union-of-subspaces dictionary learning. The same idea can be transferred to CoSpaDi by allocating $(k_\ell,s_\ell)$ per layer or group under a global budget constraint.
>
> For this reason, in the current submission we intentionally keep allocation simple and uniform in order to isolate the effect of the factorization class itself in both per-layer and grouped settings. A similar rationale applies to grouped compression: in Section 4.3.3, we follow the grouping protocol of Basis Sharing [11] for a fair like-for-like comparison with the most relevant shared-basis low-rank baseline. We agree that more adaptive grouping strategies based on weight or activation similarity are highly promising, but we view them as complementary to the main point of the paper rather than specific weaknesses of the current comparison.
>
>
>
> Overall, we thank the reviewer again for the constructive feedback. We believe the paper already addresses these points in substance, and we will tighten the presentation where helpful. The main conclusion remains unchanged: CoSpaDi introduces activation-aware sparse dictionary learning as a new and empirically effective post-training compression paradigm for LLMs, with clear advantages over standard low-rank compression under matched settings.
>
> [1] Kjersti Engan, Svein Ole Aase, and Jan H. Husoy. Method of optimal directions for frame
> design. In Proceedings of the IEEE International Conference on Acoustics, Speech, and Signal
> Processing (ICASSP), 1999.
>
> [2] Michal Aharon, Michael Elad, and Alfred M. Bruckstein. K-SVD: An algorithm for designing
> overcomplete dictionaries for sparse representation. IEEE Transactions on Signal Processing,
> 54(11):4311–4322, 2006.
>
> [3] Ron Rubinstein, Michael Zibulevsky, and Michael Elad. Efficient implementation of the K-SVD
> algorithm using Batch Orthogonal Matching Pursuit. Technical Report, Computer Science
> Department, Technion, 2008.
>
> [4] Julien Mairal, Francis Bach, Jean Ponce, and Guillermo Sapiro. Online dictionary learning
> for sparse coding. In Proceedings of the 26th International Conference on Machine Learning
> (ICML), 2009.
>
> [5] Vladimír Macko and Vladimír Boža. MACKO: Sparse Matrix-Vector Multiplication for Low
> Sparsity. arXiv preprint arXiv:2511.13061, 2025.
>
> [6] Vladimír Macko. MACKO-SpMV GitHub Repository. https://github.com/vlejd/macko_
> spmv.
>
> [7] Elias Frantar and Dan Alistarh. SparseGPT: Massive Language Models Can Be Accurately
> Pruned in One-Shot. arXiv preprint arXiv:2301.00774, 2023.
>
> [8] Mingjie Sun, Zhuang Liu, Anna Bair, and J. Zico Kolter. A Simple and Effective Pruning
> Approach for Large Language Models. arXiv preprint arXiv:2306.11695, 2023.
>
> [9] Xin Wang, Samiul Alam, Zhongwei Wan, Hui Shen, and Mi Zhang. SVD-LLM V2: Optimizing
> singular value truncation for large language model compression. In Proceedings of NAACL, 2025.
>
> [10] Qinsi Wang et al. Dobi-SVD: Differentiable SVD for LLM compression and some new perspectives. arXiv preprint arXiv:2502.02723, 2025.
>
> [11] Jingcun Wang et al. Basis Sharing: Cross-layer parameter sharing for large language model
> compression. In International Conference on Learning Representations (ICLR), 2025.

---

> > ### Comment · Reviewer_X2Ep · 2026-03-27
> >
> > Thank you for the detailed response! I don't have any further concerns.

---

### Review · Reviewer_xicA · 2026-03-21

**Summary Of Contributions:**

The paper proposes CoSpaDi, (Compression via Sparse Dictionary Learning), a training-free calibration-guided framework that replaces low-rank factorization with a structured sparse decomposition in which each weight matrix is represented as a dense dictionary multiplied by a column-sparse co-efficient matrix. This yields a union-of-subspaces representation (different columns can use different subsets of atoms), which is argued to be more expressive than low-rank at the same parameter budget.

**Audience:**

Yes

**Audience Explanation:**

It is interesting to the efficient LLM, post-training compression community.

**Claims And Evidence:**

Yes

**Claims Explanation:**

Mostly supported, especially for the quality-wise, but weak evidence for practicality / deployment efficiency.

1. The core empirical claim of better quality than SVD-based baselines at matched CR (20–40%)—is supported by multiple tables/figures in both per-layer and grouped settings.
2. The claim that data-aware calibration helps is supported by “data-free vs data-aware” comparisons (dictionary learning without whitening vs CoSpaDi with whitening).
3. What is interesting but missing is the practical speed/efficiency under a standard runtime with competitive kernels and realistic batch/seq settings.

**Requested Changes:**

- For reproducibility, please include more implementation details like dictionary initialization, OMP settings, etc.
- Please report the storage size for dictionary + values + mask/indices, vs. after applying Eq14.
- It would be interesting to measure the practical speed/efficiency under realistic batch/seq settings compared with optimized low-rank inference in the same runtime.
- Since you motivate kernels/hardware, it would be good to include at least one N:M (e.g., 2:4) or structured sparse PTQ baseline.

---

> ### Author Response · Authors · 2026-03-31
> **Response to Reviewer xicA**
>
> We thank the reviewer for his careful reading of the paper and for the constructive suggestions. We are glad that the reviewer finds the main empirical claims largely supported. We also agree that the paper would benefit from clearer reproducibility details, more explicit storage accounting, and a stronger discussion of practical deployment aspects.
>
> $\textbf{(1) Reproducibility / implementation details.}$
> We agree that the implementation details should be stated more explicitly. While we believe that the necessary details are already present in the current draft, they are currently split between the main text and the appendix, which makes them harder to locate. In particular, for the main results we use K-SVD with power-iteration-based atom updates with $n_{power}$ = 8, and for the sparse coding stage we use batched OMP with mini-batch size 8192. The dictionary is initialized by a random permutation of columns of the original projection matrix. Full list of used hyperparameters are listed in the Table 1. We agree that these details should be collected more clearly in one place. We also emphasize that reproducibility is supported not only by the paper text, but also by the code release. In particular, the full source code has already been attached as supplementary material to the submission. We will make this more explicit in the revision, so that readers can more easily identify the code package and use it to reproduce the reported results.
>
>
> **Table 1.** Hyperparameters used in all experiments unless otherwise specified.
> | Hyperparameter | Value |
> |---|---|
> | Random seed | 42 |
> | K-SVD iterations ($n_\text{iter}$) | 60 |
> | $k/s$ ratio ($r_{ks}$) | 2.0 |
> | SVD solver | power |
> | Power iterations ($p_\text{iter}$) | 8 |
> | Ridge regularization ($\lambda$) | $10^{-6}$ |
> | OMP batch-size | 8192 |
> | Target modules | linear layers in transformer blocks |
> | Dictionary size $k$ | $\frac{(1 - \text{CR}) d_1 d_2}{d_1 + d_2 / \rho}$ |
> | Sparsity $s$ | $k / \rho$ |
> | Dictionary init | random columns of W |
> | Calibration dataset | RefinedWeb |
> | Number of samples | 256 |
> | Sequence length | 1024 |
>
> In the revision, we will add a dedicated paragraph on implementation details and reproducibility that consolidates the information already present in the paper, including:
>  - the exact solver configuration used for the main results,
>  - dictionary initialization,
>  - sparse-coding details,
>  - the number of alternating steps and power iterations,
>  - calibration setup,
>  - the exact storage convention used for the reported compression ratios,
>  - an explicit note that the full implementation is provided in the supplementary material.

---

> ### Author Response · Authors · 2026-03-31
> **Response to Reviewer xicA**
>
> **(2) Storage accounting: dictionary + values + mask/indices vs. Eq. 14.**
>
> We agree that this distinction should be made more transparent. The current paper already gives the full storage accounting including dictionary, stored coefficient values, and packed binary mask, and also discusses the simplified accounting used after coefficient truncation, which leads to Eq. 14. However, we agree that the presentation can be clearer and more explicit.
>
> To address this point directly, in the revision we will include a compact storage-breakdown table of the following form for the main settings:
>
> **Table 2: Memory footprint breakdown (MB) for storing all linear layers (excluding embedding, lm_head, normalization layers) under the CoSpaDi decomposition at various compression rates.**
>
> | Model | Applied eq. | Quant. coeff. | Dictionary | Coeff. values | Mask/indices | Full total |
> |---|---|---|---|---|---|---|
> | Llama-3.2-1B | - | No | - | - | - | 1856 |
> | CoSpaDi, CR=20% | Eq. 12 | No | 990 | 438 | 54 | 1484 |
> | | Eq. 14 | Yes | 1018 | 408 | 58 | 1484 |
> | CoSpaDi, CR=30% | Eq. 12 | No | 866 | 383 | 47 | 1298 |
> | | Eq. 14 | Yes | 890 | 357 | 51 | 1298 |
> | CoSpaDi, CR=40% | Eq. 12 | No | 743 | 329 | 41 | 1113 |
> | | Eq. 14 | Yes | 763 | 306 | 43 | 1113 |
> | Qwen-3-8B | - | No | - | - | - | 13248 |
> | CoSpaDi, CR=20% | Eq. 12 | No | 7005 | 3192 | 399 | 10596 |
> | | Eq. 14 | Yes | 7226 | 2948 | 421 | 10596 |
> | CoSpaDi, CR=30% | Eq. 12 | No | 6129 | 2793 | 349 | 9272 |
> | | Eq. 14 | Yes | 6322 | 2579 | 368 | 9271 |
> | CoSpaDi, CR=40% | Eq. 12 | No | 5253 | 2393 | 299 | 7946 |
> | | Eq. 14 | Yes | 5419 | 2211 | 315 | 7947 |
>
> Columns report the storage required for dictionaries, coefficient values, and mask/index overhead. The "Quant. coeff." and "Applied eq." columns indicate whether coefficient truncation is applied (Eq. 14) or not (Eq. 12). Note that while the component-wise allocation shifts, the full total remains unchanged.
>
>
> **Concrete Example.** For clarity, we walk through the storage cost for a single down_proj
> matrix in Qwen-3 8B, where d1 = 4096 and d2 = 12,288.
>
> **Using Eq. 12.** The dictionary has shape (4096 × 3657), requiring ≈ 28.57 MB. The coefficient
> matrix has shape (3657 × 12288) with 1828 nonzero elements per column, yielding ≈ 42.84 MB for
> the coefficients and 5.36 MB for the binary mask.
>
> **Using Eq. 14.** The dictionary has shape (4096 × 3932), requiring ≈ 30.72 MB. The coefficient
> matrix has shape (3932 × 12288) with 1966 nonzero elements per column, yielding ≈ 40.32 MB for
> the coefficients (truncated to 14-bit precision) and 5.76 MB for the binary mask.
>
> This should make the storage comparison fully transparent and clarify precisely how much each
> component contributes to the total compressed size under both accounting conventions.

---

> ### Author Response · Authors · 2026-03-31
> **Response to Reviewer xicA**
>
> **(3) Practical speed / efficiency under realistic batch and sequence settings.**
>
> We thank the reviewer for raising this point. We agree that practical deployment efficiency should be distinguished carefully from asymptotic complexity. In fact, this distinction is already part of the current paper: we compare CoSpaDi against low-rank factorization not only at the level of parameter budget and asymptotic multiplication cost, but also in terms of measured inference behavior (see figures 5-7 in Appendix). Our current results show that CoSpaDi is broadly on par with low-rank methods from the practical speed perspective, rather than being clearly worse.
>
> Additionally, following the evaluation protocol used in the SVD-LLM repository, we measured end-to-end inference throughput in tokens-per-second for Llama-3 8B and Qwen3-8B with generated length of 1024 across different batch sizes. The results show that, although factorized projection layers can reduce the effective compute cost, the overall model-level throughput does not consistently improve for either SVD or CoSpaDi. This suggests that bottlenecks beyond the decomposed linear layers dominate end-to-end latency in the current runtime stack. Importantly, within this setup CoSpaDi remains broadly comparable to SVD-based compression in throughput.
>
> **Table 3: End-to-end throughput (tokens/s) for Llama-3 8B using SVD and CoSpaDi under the evaluation protocol of the SVD-LLM repository.**
>
> | Compression | BS=4 | BS=16 | BS=32 |
> |---|---|---|---|
> | Original | 65.6 | 465.5 | 660.0 |
> | SVD (CR=0.5) | 50.9 | 204.0 | 706.4 |
> | SVD (CR=0.4) | 49.3 | 184.5 | 496.7 |
> | SVD (CR=0.3) | 50.5 | 191.9 | 573.6 |
> | SVD (CR=0.2) | 50.1 | 362.1 | 569.6 |
> | CoSpaDi (CR=0.5) | 51.5 | 349.2 | 583.7 |
> | CoSpaDi (CR=0.4) | 49.8 | 184.4 | 462.8 |
> | CoSpaDi (CR=0.3) | 48.9 | 178.3 | 394.9 |
> | CoSpaDi (CR=0.2) | 50.0 | 185.7 | 481.0 |
>
> **Table 4: End-to-end throughput (tokens/s) for Qwen3-8B using SVD and CoSpaDi under the evaluation protocol of the SVD-LLM repository.**
>
> | Compression | BS=4 | BS=16 | BS=32 |
> |---|---|---|---|
> | Original | 46.9 | 179.3 | 552.4 |
> | SVD (CR=0.5) | 37.8 | 304.6 | 504.4 |
> | SVD (CR=0.4) | 36.7 | 271.7 | 429.7 |
> | SVD (CR=0.3) | 37.3 | 281.2 | 453.6 |
> | SVD (CR=0.2) | 36.7 | 263.2 | 401.6 |
> | CoSpaDi (CR=0.5) | 37.3 | 249.5 | 404.4 |
> | CoSpaDi (CR=0.4) | 36.7 | 267.4 | 404.6 |
> | CoSpaDi (CR=0.3) | 36.6 | 275.0 | 430.8 |
> | CoSpaDi (CR=0.2) | 37.1 | 286.0 | 455.6 |
>
> We also note that CoSpaDi should be viewed in connection with recent progress in sparse inference at moderate sparsity levels. In our main setting, the default ratio ρ = k/s = 2 corresponds to 50% sparsity in the coefficient matrix. This regime is especially relevant because recent work such as MACKO reports promising sparse matrix-vector multiplication speedups already at such moderate sparsity levels, suggesting that the practical systems outlook for methods like CoSpaDi may improve as sparse kernels continue to mature [1]. At the same time, we emphasize that translating projection-layer FLOP reductions into consistent full-model throughput gains still requires runtime and kernel-level support, which is outside the main scope of the present paper.
>
> Recent work on LLM compression confirms that unstructured sparsity can yield practical inference speedups with existing engines. For instance, ELSA [2] reports CPU inference speedup using DeepSparse [3] and GPU inference speedup using nm-vllm [4] for unstructured sparse matrix multipication. Since CoSpaDi’s inference involves a dense-times-sparse multiplication, the same acceleration engines are directly applicable to our method.
>
>
> [1] Vladimír Macko and Vladimír Boža. MACKO: Sparse Matrix-Vector Multiplication for Low
> Sparsity. arXiv preprint arXiv:2511.13061, 2025.
>
> [2] Haoxian Chen and Likang Wu and Ming He and Jianping Fan and Limin Wang. Ef-
> ficient Low-rank and Sparse Approximation and Adaptation for Large Language Models
> https://openreview.net/forum?id=FcQdppbqDo, 2025.
>
> [3] Neural Magic. DeepSparse https://github.com/neuralmagic/deepsparse
>
> [4] Neural Magic. nm-vllm https://github.com/neuralmagic/nm-vllm

---

> ### Author Response · Authors · 2026-03-31
> **Response to Reviewer xicA**
>
> **(4) Semi-Structured sparse N:M baseline.**
>
> We thank the reviewer for his helpful suggestion. To address the reviewer’s comment we have performed a comparison against a recent training-free unstructured/semi-structured pruning strategy [5] (WANDA), which supports N:M sparsity patterns. In this context, it is worth noting that while 2:4 sparsity, which we consider in our comparison, nominally zeroes out 50% of the weights, the effective storage compression is less than 2× due to metadata overhead. Specifically, in the standard NVIDIA semi-structured format, each non-zero element requires 2 bits of metadata, yielding 9 bits per element for 16-bit weights (compared to 16 bits dense) and an effective compression ratio of 1 - 9/16 = 43.8% of the original size rather than the 50% one might naively expect. Based on the results in Table 5 we observe that Cospadi is still very competitive to recent semi-structured methods in terms of performance, providing the best results at the same compression ratio.
>
> It is also worth mentioning that our framework can also accommodate the same N:M sparsity patterns in the sparse coefficient matrix S. This would amount to utilizing different number of atoms from the dictionary to approximate each column of the original model weight, instead of the fixed number of atoms that we are currently using. This is an interesting direction that we plan to investigate in the near future.
>
> **Table: Performance comparison with Wanda for Llama3 8B**
>
> | Method | CR | PIQA | HellaSwag | LAMBADA | ARC-e | ARC-c | SciQ | Race | MMLU | Avg. | WikiText (PPL↓) | LAMBADA (PPL↓) |
> |---|---|---|---|---|---|---|---|---|---|---|---|---|
> | Llama3 8B | -- | 80.7 | 79.1 | 75.6 | 77.7 | 53.5 | 93.9 | 40.3 | 62.2 | 70.4 | 7.3E+00 | 3.1E+00 |
> | SVD-LLM | 0.4 | 60.3 | 34.5 | 11.4 | 32.4 | 24.5 | 44.2 | 25.7 | 23.1 | 32.0 | 5.5E+02 | 1.3E+03 |
> | CoSpaDi | 0.4 | 63.7 | 41.4 | 30.3 | 39.1 | 26.6 | 68.5 | 30.5 | 25.4 | 40.7 | 1.8E+02 | 1.2E+02 |
> | Wanda 2:4 | 0.4 | 62.4 | 39.7 | 0.8 | 41.9 | 25.6 | 74.7 | 26.8 | 23.4 | 36.9 | 2.1E+02 | 5.7E+03 |
>
>
> Overall, we thank the reviewer again for these suggestions. We believe they will help improve the clarity of the paper, especially with respect to reproducibility, storage transparency, and deployment-oriented positioning.
>
>
>
> [5] Mingjie Sun and Zhuang Liu and Anna Bair and J Zico Kolter A Simple and Effective Pruning
> Approach for Large Language Models In The Twelfth International Conference on Learning
> Representations (ICLR), 2024.

---

### Decision · Action_Editor_wbw6 · 2026-05-08

**Recommendation:** Reject

**Additional Comments:**

A resubmission should provide clear and convincing evidence for any claims across a more representative selection of existing efficient inference baselines, with comparisons both on real-world throughput and the effect on performance on a range of benchmarks.

**Audience:**

No

**Audience Explanation:**

I believe a novel methodology for efficient inference in LLMs that lacks comparison in real-world timings convincingly demonstrating improvements to throughput and/or latency as compared to a representative selection of existing efficient inference methods would find little interest within the efficient LLM inference research community.

**Claims And Evidence:**

No

**Claims Explanation:**

The reviewers found a lot to like about the work, and were generally satisfied with the correctness of most claims in the paper and rebuttal as to the implementation, reproducibility, the benchmark evaluation, and scalability of the methodology to find the representation. Notably, the methodology appears to preserve quality of the representation relatively well with compression ratio across a wide range of benchmarks (although with limited comparison to baselines), and to be novel in the specific context presented. The authors notably made significant improvements to the work through rebuttal to address most of the reviewer's initial concerns.

However, multiple reviewers had significant concerns that claims on inference efficiency are not supported by clear and convincing evidence, beyond theoretical computational complexity, and I believe the rebuttal did not sufficiently address these concerns. For a work that motivates itself primarily by efficiency, computational complexity is clearly not enough. As the authors themselves explain well on the topic of unstructured sparsity in their background, performance on real-world hardware often does not match theoretical gains, and only strong evidence in real-world timings compared to a representative selection of existing efficient inference baselines can support such claims on real-world efficiency vs accuracy. Notably, much of the author's discussion in response to reviewer concerns on real-world inference timings related their work to (weight) sparsity, and work/libraries on accelerating inference for those representations. The authors claim that these acceleration engines would be directly applicable to CoSpaDi, while also not further discussing why they did not use these to show inference timings in their work.

There are some results on the inference time vs. compression ratio for Llama-3 models on a A100 in Fig 5 and 6, but these are only for specific layers, and lack any comparison with other efficient representations. During rebuttal the authors added throughput comparisons in Appendix A.10 with a single low-rank representation (SVD-LLM). CoSpaDi only appears to match the throughput of that particular baseline, with the claim being that accuracy is better maintained for a given throughput, a potentially interesting finding, however this single baseline is not enough to judge if this claim holds in general. Finally, while comparing compression ratios and accuracy of CoSpaDi relative to popular pruning-based approaches in Appendix A.11, there is surprisingly no comparison with the relative latency or throughput of those sparse approaches.

There is a very large range of literature on efficient representations (e.g. low-rank, sparse and quantized) missing in the paper which would appear to be relevant as baselines for comparison, not only to support the claims of the approach, but also for the reader to understand where CoSpaDi has a benefit over (or even in tandem with) existing approaches. Further, as pointed out by the reviewers, the method itself uses a relatively simple uniform compression budget across layers, while many other these baselines do not. The authors claim this is orthogonal to the effectiveness of CoSpaDi, but it would rather appear necessary to fairly compare with existing methodology which often finds its best results/strength under dynamic compression, whereas it's not clear if CoSpaDi has a similar property.

Given the promising initial findings and general interest the reviewers found in the work otherwise, especially post-rebuttal, I would encourage the authors to work further to address these gaps. I am recommending allowing a resubmission if these appear to be adequately addressed.

**Resubmission Of Major Revision:**

The authors may consider submitting a major revision at a later time.